# Inhibitory activity of traditional plants against *Mycobacterium smegmatis* and their action on Filamenting temperature sensitive mutant Z (FtsZ)—A cell division protein

**Radhika Ravindran**, **Gayathri Chakrapani, Kartik Mitra, Mukesh Doble**\*

Bioengineering and Drug Design Lab, Department of Biotechnology, Indian Institute of Technology, Madras, India

\* mukeshd@iitm.ac.in

**Data Availability Statement:** All relevant data are within the manuscript and its Supporting Information files.

## Abstract

The study was designed to assess whether plant extracts / phytochemical (D-Pinitol) synergistically combine with antituberculosis drugs and act on *Mycobacterium smegmatis* (*M. smegmatis*) as well as assess their mode of action on *Mycobacterium tuberculosis* (*M.tb*) Filamenting temperature sensitive mutant Z (FtsZ) protein. Resazurin microtitre plate assay (Checker board) was performed to analyze the activity of plant extracts against *M. smegmatis*. Synergistic behaviour of plant extracts / D-Pinitol with Isoniazid (INH) and Rifampicin (RIF) were determined by time–kill and checker board assays. Elongation of *M. smegmatis* cells due to this treatment was determined by light microscopy. The effect of Hexane methanol extract (HXM) plant extracts on cell viability was determined using PI/SYTO9 dual dye reporter Live/Dead assay. Action of HXM plant extracts / D-Pinitol on inhibition of FtsZ protein was done using Guanosine triphosphatase (GTPase) light scattering assay and quantitative Polymerase Chain Reaction (qPCR). The Hexane-methanolic plant extract of *Acacia nilotica*, *Aegle marmelos* and *Glycyrrhiza glabra* showed antimycobacterial activity at 1.56 ± 0.03, 1.32 ± 0.02 and 1.25 ± 0.03 mg/mL respectively and that of INH and RIF were 4.00 ± 0.06 μg/mL and 2.00 ± 0.04 μg/mL respectively. These plant extracts and major phytochemical exudate D-Pinitol was found to act synergistically with antimycobacterial drugs INH and RIF with an FIC index ~ 0.20. Time-Kill kinetics studies indicate that, these plant extracts were bacteriostatic in nature. D-Pinitol in conjunction with INH and RIF exhibited a 2 Log reduction in the growth of viable cells compared to untreated. Attempt to elucidate their mode of action through phenotypic analysis indicated that these plant extracts and D-Pinitol was found to interfere in cell division there by leading to an abnormal elongated cellular morphology. HXM extracts and D-Pinitol synergistically combined with the first line tuberculosis drugs, INH and RIF, to act on *M. smegmatis*. The increase in the length of *M. smegmatis* cells on treatment with D-Pinitol and HXM extract of the plants indicated that they hinder the cell division mechanism thereby leading to a filamentous phenotype, and finally leading to cell death. In addition, the integrity of the bacterial cell membrane is also altered causing cell death. Further gene expression analysis showed that these plant extracts and D-Pinitol hampers

**Funding:** Funding is not received.

**Competing interests:** The authors have declared that no competing interests exist.

with function of FtsZ protein which was confirmed through *in vitro* inhibition of FtsZ–GTPase enzymatic activity.

## 1. Introduction

*Mycobacterium tuberculosis* is a pathogenic organism which causes Tuberculosis. About a quarter of the global population is affected with this disease [1]. Due to emerging drug resistant strains, and reduced effectiveness of treatment due to failure in patient adherence to treatment regime leads to complication and failure of treatment [2]. Therefore, there is a need to screen for novel antimycobacterial medicinal plant extracts to employ them as complementary and adjuvant medicine along with the conventional chemotherapy to increase the effectiveness and action of chemotherapeutic drugs. Traditionally, plant extracts and their active components have been used to treat many diseases, and the structures of many phytochemicals have been the starting scaffold for the design of synthetic drugs, including aspirin and taxol [3]. Plant extracts possess phenolic compounds and their derivatives play an important role to protect the human body against the damage caused by free radicals [4].

Many plant extracts and compounds were tested against mycobacteria and few were reported for their antituberculosis activity. Chloroform extracts of *Pterolobium stellatum* (Forssk), *Persea americana* Mill L and *Otostegia integrifolia* Benth L have shown Minimum Inhibitory Concentration (MIC) values of 0.312, 2.5 and 0.312 mg/mL respectively against *M. tuberculosis* strain H37R$_V$ [5]. Methanolic extract of *Aegle marmelos* L, *Glycyrrhiza glabra* L, *Lawsonia inermis* L, *Piper nigrum* L and *Syzygium aromaticum* L, exhibited antituberculosis activity at a range of 0.8 to 100 μg/mL against *M. tuberculosis* strain H37R$_V$ [6]. While ethyl acetate extract of *Piper longum* L inhibited *M. smegmatis* at 32 mg/mL [7]. Ethanolic extracts of *Boswellia serrata* Roxb.ex, *Datura stramonium* L and *Lavandula stoechas* L inhibited *M. tuberculosis* strain H37R$_V$ with a MIC in the range of 125 to 250 μg/mL [8]. Phytochemicals namely, Distemonanthoside, 4-Methoxygallic acid, Quercetin and Sitosterol 3-*O*-β-D-gluco-pyranoside inhibited *M. tuberculosis* strain H37R$_V$ with a MIC at a range of 31 to 125 μg/mL [9]. Oleanolic acid declined the growth and development of *M. tuberculosis* strain H37R$_V$ at a MIC of 50–200 μg/mL [10].

FtsZ protein is a bacterial tubulin homolog involved in the creation of a Z-ring at the site of cell division. FtsZ is a Guanosine TriPhosphate (GTP) / Guanosine DiPhosphate (GDP) binding protein with the ability of polymerising GTP-into protofilaments. Abnormalities in polymerization / GTPase activity will lead to the inhibition of Z-ring which makes the cell elongated and finally leads to the death of an organism [11]. This crucial behavior of protein motivated many researchers around the world to focus and design novel inhibitors targeting it. Berberine, chrysophaentins A-H, Cinnamaldehyde, Curcumin and Viriditoxin are potent inhibitors that are known to target GTPase activity of FtsZ [12].

Antimycobacterial activity of HXM extracts of three plants namely *Acacia nilotica*, *Aegle marmelos* and *Glycyrrhiza glabra* were studied. *Acacia nilotica*, L *(A. nilotica)* belongs to the family of Fabaceae commonly known as Babul, Karuvelam or Kikar. This plant is distributed in all parts of the world. It is used extensively for the treatment of various types of cancers like bone, mouth and skin by traditional healers in different regions of Chattisgarh (India). In West Africa, the root of *A.nilotica* is used to treat tuberculosis, the wood is used to treat small-pox and the leaves are used to treat ulcers [13]. *A. nilotica* extract is used traditionally to treat respiratory related diseases. It has antituberculosis effects and it could serve as lead for developing new antibiotics [14]. Traditionally, the plant roots, leaves, flowers, buds has anticancer, antimicrobial, antioxidant, antimutagenic activity and is used for treating cough, dysentery,

leprsosy, opthalmia, small pox, skin ulcers and tuberculosis as well as astringent, antispasmodic and aphrodisiac [15] and [16].

*Aegle marmelos* L, *(A.marmelos)* belongs to the Rutaceae family and is also called as bael tree. It is used in indigenous system of Indian medicine. It is native to India and grows wild in Sub-Himalaya from Jhelum, eastwards towards west Bengal, in central and south India [17]. It is known to possess antidiabetic, anticancer, antiinflammatory and antimicrobial activity [18]. Fruit of *A. marmelos* is also used in the treatment of asthma, dyspepsia, diarrhea, digestive, dysentery, hepatitis, sinusitis, tuberculosis and stomachic [19]. Compounds isolated from fruit extract of *A.marmelos* like coumarins, marmelosin, marmin, xanthotoxol, kaempferol 3 O-rhamnoside and afzelin have been tested against *Mycobacterium tuberculosis* and *Mycobacterium bovis*. Coumarins and marmelosin have shown antimycobacterium activity against *M. tuberculosis* H37Rv with an $IC_{50}$ value of 12.46 μg/mL and 4.31 μg/mL [20].

*Glycyrrhiza glabra* L *(G. glabra)*, belongs to the family Fabaceae. It is also called as Licorice. Rhizome and root of *G. glabra* are used as carminative and expectorant by the Chinese, Egyptian, Greek, Indian and Roman civilizations. It is also called as sweet wood and is native to the certain areas of Asia and Mediterranean. It is an old age plant used in traditional medicine for its ethanopharmacological use to cure simple cough to complex cancer and SARS virus [21]. Rhizomes and roots are used orally to treat addison disease, diabetes, lung ailments, kidney stones and tuberculosis. It is also used as mild laxative, contraceptive and to improve sexual function [22]. The roots of *G. glabra* is used traditionally in India to treat pulmonary related diseases, chest ailments, persistent cough and manage tuberculosis [23]. Licorice also shows antiplatelet aggregation effects [24] and it has the ability to relieve cough from ancient times [25].

D-Pinitol, a phytochemical which is predominantly present in all the three HXM extracts (GCMS & HPLC) is studied in detail here. It has the ability to reduce metastasis of human lung cancer [26], it has anti-inflammatory, antihyperlipdemic, cardioprotective and antioxidant activities, [27] and [28]. It is used for cancers of lung, bladder and breast [29] and [30]. It is also effective in the inhibition and progression of prostate cancer [31] and hypoglycemic levels [32].

Initially, antimycobacterial screening of HXM extracts of selected plants were carried out, and the extracts that showed activity were further characterized by Gas Chromatography Mass Spectrophotometer (GCMS) and High Performance Liquid Chromatography (HPLC). The HXM extracts were tested individually as well as in combination with two current antimycobacterial drugs namely INH and RIF by determining the MIC and potential synergy. Time-kill studies were done to determine whether the extracts possess bactericidal or bacteriostatic behaviour. Morphology of cells was measured by the cell elongation study. Antioxidant activity of the plant extracts was also studied. The mechanism of action of these HXM extracts / compound were determined by Live/dead assay. The effect of HXM extracts / compound on *M. smegmatis* FtsZ were also investigated by GTPase light scattering and q PCR studies.

## 2. Materials and methods

### 2.1. Bacterial strain

*Mycobacterium smegmatis* strain (ATCC14468) was procured from the Institute of Microbial Technology, (IMTECH), Chandigarh, India. The bacteria were cultured and maintained at a pH 7, temperature at 37°C in Middlebrook 7H9 medium, supplemented with OADC (Oleic acid/bovine Albumin, Dextrose, Catalase). These chemicals were procured from Sigma (Aldrich, St. Louis, USA). OADC (10%) was prepared separately using, 0.06 mL of oleic acid, 5 g of albumin, 0.85 g of sodium chloride, 2 g of dextrose, 3 mg of catalase and

100 mL of filter sterilized distilled water. The mixture was filter sterilized using 0.44μm membrane. It was added separately to the culture medium at 2% concentration (v/v).

## 2.2 Plant collection

Leaves of *A. nilotica* were collected from IIT Madras (South India) Campus. Fruits of *A. marmelos* (Bael fruit) were collected from a tree near the campus. No specific permission was required for collection of plant parts from the IIT Madras (South India) authorities as they are very common and abundant species in the region. The roots of *G. glabra* were purchased from the Maha Raja Herbals, an ayurvedic shop in Chennai (South India). All the three plant species used in the study are neither endangered nor protected. Collected plant samples were identified as leaves of *A. nilotica*, fruits of *A. marmelos* and roots of *G. glabra* by Dr. K.N.Sunil Kumar, Research Officer, HOD Pharmacognosy, Siddha Central Research Institute, Arumbakkam, Chennai, 600106. The samples were deposited in the Herbarium of Siddha Central Research Institute, Arumbakkam, Chennai, 600106. The voucher specimens of the above samples are A14061903N, A14061901M, G14061902G respectively.

## 2.3. Herbal preparation

The leaves of *A. nilotica* were thoroughly washed, rinsed with distilled water and dried overnight in a hot air oven at 37˚C. Unripened pulp of *A. marmelos* was well macerated and lyophilized into a fine powder. Roots of *G. glabra* were broken into small pieces, washed, rinsed with distilled water and dried overnight in a hot air oven and powdered using a hand blender.

## 2.4. Extraction procedure

20 g of each of the powdered leaves of *A. nilotica*, fruit of *A. marmelos* and roots of *G. glabra* were weighed separately. Each plant powdered sample was defatted with hexane (300 mL) and the mixtures were kept overnight at room temperature with occasional stirring. The hexane treated residues of the above plant parts were filtered separately to remove the fatty volatile mass and partitioned with methanol (150 mL) overnight. Methanol soluble extracts were filtered out, concentrated using a rotavapor in a preweighed flask. This is named as HXM extract.

## 2.5. Chemicals

Chemicals and reagents namely Dimethyl sulfoxide (DMSO), 2,2-diphenyl-1-picrylhydrazyl (DPPH), D-Pinitol, GTP, INH, Isopropyl β-D-1-thiogalactopyranoside (IPTG) and RIF were procured from Sigma (Aldrich, St. Louis, USA).

## 2.6. Antimycobacterial agents

Antimycobacterial activity of the HXM extracts were compared with two standard antitubercular drugs namely, INH and RIF. Stock solution of INH (1mg/mL) was prepared with sterile distilled water and RIF (1mg/mL) was prepared with DMSO. 100 μl from the stock concentration was made up to 1mL with the corresponding DMSO/ water. Such that maximum concentration of 100 μg/mL was used for the assay.

## 2.7. High Performance Liquid Chromatography (HPLC)

A Phenomenex Rezex R-OA organic acid column (300 × 7.8 mm) fitted on a Shimadzu Prominence HPLC and a RI (refractive index) detector were used for the analysis. Elution was carried out at 50˚C using 0.6 mL / min of 5 mM $H_2SO_4$. Internal diameter of the HPLC column is

7.8 mm The Shimadzu LC solution software was used to estimate the peak of plant extracts and D-Pinitol Injection volume of the sample was 20 μl. HPLC was followed according to the method as described [33,34].

## 2.8. Preparation of D-Pinitol standard solution

Standard solution was prepared by dissolving 1mg of D-Pinitol in 1 mL of HPLC grade water and filtering it through 0.45 μm filter paper.

## 2.9. Gas chromatography and mass spectroscopy (GCMS)

The phytochemical investigation of HXM extracts were performed on a GC-MS, Clarus 500 Gas Chromatograph Perkin Elmer, USA. Capillary standard non-polar column measured dimension was 30Mts. Temperature of the oven was set to 50°C while injection temperature was maintained at 250 °C and injection volume was 1 μL. Plant extracts were rederivatized using BSTFA (*N*,*O*-Bis(trimethylsilyl)trifluoroacetamide).

## 2.10. Resazurin assay (REMA)

Resazurin microtitre assay was performed as per the reported protocol [35–37]. To prepare the inoculum, two loops of the bacteria were suspended in 15 mL of Middlebrook 7H9 medium in sterile vial and the culture was incubated in orbital shaker at 180 rpm. The cultures were taken for the study once it reached an O.D of 0.6–0.7 (JASCO UV Spectrophotomter, 600 nm). Middlebrook 7H9 broth supplemented with OADC (10%) of 100 μL was added to each flat well bottom of a 96 well microtitre plate. Serial two-fold dilutions of three HXM extracts (dissolved in 5% DMSO), D-Pinitol were performed in Middlebrook 7H9 broth to obtain concentration range of 0.75 mg/mL and 100 mg/mL respectively. Concentration of INH and RIF used is 100 μg/mL serves as positive controls. Culture medium alone and culture medium with *M. smegmatis* were the blank and negative control. Controls like DMSO and 5% DMSO were also used for the study. *M.smegmatis* suspension (100 μL) containing approximately $1 \times 10^6$ CFU/mL was also added to all the wells containing the samples to yield a final volume of 200 μL/well. The plate was wrapped using aluminium foil and incubated at 37°C for 48 hours. After incubation, Resazurin dissolved in water (0.2 mg/mL) of 20 μL was added to each well and wrapped again in aluminium foil and incubated for 4–5 hours. The change in color of the solution from blue to pink indicates bacterial growth and blue indicates no growth.

## 2.11. Combinatorial studies between plant extracts and drugs

Checkerboard broth microdilution method was performed to find the effect of combination of drugs and plant extracts / phytochemical. Briefly, drugs (INH / RIF) are serially diluted vertically and different concentration of plant extract has been added horizontally in such a way that each concentration of plant extract has been tested in combination with decreasing concentration of the drugs. Middlebrook 7H9 broth of 100 μL was added into all the wells. Two fold serial dilution of drugs (INH / RIF) has been done vertically and 50 μL aliquots of different concentration of HXM plant extracts / compound were arranged for every conjunction to be tested. 100 μL of *M.smegmatis* cultures were added uniformly and incubated for 24 hours at 37°C. After the incubation, Resazurin dye of 20 μL (0.02%) was added to all the wells and plates were further incubated for 5 hours. Lowest concentration of antibiotic and plant extracts in combination showing no color change is considered as measure of inhibition of bacterial growth. Synergistic interactions between plant extracts and drugs were calculated by estimating the FIC index, using the formula, (MIC of antibiotic in combination / MIC of antibiotic

alone) + (MIC of plant extract in combination / MIC of plant extract alone) [38]. The combination is considered synergistic, additive and antagonist based on the FIC index as < 0.5, 1 or > 1.5 respectively.

## 2.12. Time kill assay

Combinations of drugs with extracts were employed equivalent to 1 x MIC. 10 mL of treatment mixtures were arranged and inoculated with 20 μL of 2-day old culture to reach a final Optical Density (O.D) of 0.6 (i.e. $2 \times 10^7$ cells). 50 μL aliquots were withdrawn at 0, 1, 2, 3, 4, 5, 6 and 7 days, diluted serially and corresponding dilution was spread plated on 7H10 agar and incubated at 37˚C for 48 hours. Bacterial cell colonies were enumerated by help of magnifying lens, and a kill curve response has been plotted between time and $Log_{10}$ Colony Forming Unit (CFU) to determine the mode of action of the combination [39].

## 2.13. Bacteria live / dead assay

Live / Dead assay was performed as per manufacturers protocol using Baclight kit method [40]. Briefly, the bacterial cultures were grown at 37˚C and centrifugation done at 10,000 rpm for 10 minutes at 4˚C and the supernatant collected were discarded. Pellets were washed in sterile water. HXM extracts, D-Pinitol, INH / RIF were added at 1 x MIC and this setup was kept at 37˚C for 4 hr. After incubation, equal amount of Bac light reagent was added, the vials were covered in aluminum foil and incubated at 37˚C for 15 minutes. After this, around 2–5 μL of above samples were mounted on the microslide for viewing the dead and live cells using 40x objective using a Carl-Zeiss Fluorescence Microscope (Axio Imager 2 for Life Science Research, Germany). The kit consists of SYTO9 (green fluorescence dye) is read at 480/500 nm and propidium iodide (red fluorescence dye) is read at 490/635 nm. The former stains the bacteria with both intact as well as damaged membranes, and the latter stains only the cells with damaged membranes.

## 2.14. Cell elongation studies

The effect of above plant extracts and D-Pinitol on the increase in the *M. smegmatis* cells was determined. Briefly, overnight *M. smegmatis* grown culture (0.6–0.8 O.D) were treated separately with plant extracts and D-Pinitol (1 x MIC) and incubated for 4 hours at 37˚C. Centrifugation was done for treated samples at 10,000 rpm for 10 minutes at 4˚C. Supernatant collected was removed and the pellets were washed in sterile Phosphate buffered saline (PBS). This process is repeated two times. About 0.1–0.2 μL of the washed cell pellets were smeared on a clean microscopy slide. Cell morphology was observed at 40x objective under a Carl-Zeiss Bright Field Microscopy (Axio Imager 2 for Life Science Research, Germany) and the elongated cell length was measured using Image J software.

## 2.15. Inhibition of GTPase

pSAR1 plasmid harbouring *M. tuberculosis* FtsZ gene construct in pET15 vector was a generous gift from Dr.Malini Rajagopalan, University of Texas, USA. Plasmid was expressed in BL21DE3 *E.coli* strain exposed to IPTG (1mM) and purified by Ni-NTA column chromatography. Expression and purification of the protein was carried according to method described [41] and the purity was checked by SDS-PAGE analysis. Inhibition of GTPase was studied according to the method described [12]. Briefly, hydrolysis reaction was carried out using FtsZ (12 μM) and GTP (1 mM) in 50 μL reaction buffer with / without HXM extracts and D- Pinitol at a concentration of 1.5 x MIC. They are incubated for 30 minutes at 37˚C. Post 30 minutes

0.5 mM Ethylenediaminetetraacetic acid (EDTA) was added to quench the reactions and aliquots of about 5 μL was drawn and 120 μL of malachite green reagent was added and incubated again for 5 minutes at room temperature. This was followed by the addition of 30 μL of 34% Sodium Citrate. Absorbance was read against 630 nm using Enspire plate reader (Perkin-Elmer LS-55, Sparta, NJ) and the obtained results were compared with the standard graph of GTPase against $KH_2PO_4$.

### 2.16. FtsZ polymerization assay

FtsZ polymerization results in the formation of protofilament and monitoring the protofilament with a beam of monochromatic light causes scattering of the rays [42]. Right angle (90˚) light scattering assay was used to study the ability to disrupt the protofilament formation by the selected plant extracts / D- Pinitol at 1.5 x MIC. This was monitored with JASCO Spectrofluorometer with both excitation and emission set at 400 nm. The reactions were carried out using a polymerization buffer (50 mM MoPs, 100 mM $MgCl_2$ & 50 mM KCl). Mtb-FtsZ (12 μM) is first incubated with / without plant extracts and D- Pinitol and monitored for 300 seconds to establish the baseline (zero condition). Then, 50 seconds later 1 mM GTP was quickly added by interrupting the measurement and the scatter intensity is measured continuously for 1000 seconds. For all treatments, the time lag addition of GTP was maintained between 5 and 6 seconds [12,43].

### 2.17. FtsZ gene expression study

To ascertain the effect of the HXM extracts / compound on the gene expression of FtsZ, real time-PCR was done as explained by [44] with some variations. The housekeeping gene, 16S rRNA rrsB of *M. tuberculosis* was used as reference and using this FtsZ gene fold expression was computed by $2^{-(\Delta\Delta Ct)}$ model [45]. *M. smegmatis* culture at O.D of 0.1 was treated with HXM extracts / compound at 1.5 x MIC [46] for 4 hours [47] [48] and RNA isolation was done by Trizol Reagent method [49]. Isolated messenger Ribonucleic acid (mRNA) was transformed to its corresponding complementary deoxyribonucleic acid (cDNA) using cDNA synthesis kit (Takara Bio Inc) and quantified with qPCR, by employing SYBR green reporter. Molecular Probe to obtain the cycle threshold-$C_t$ values of untreated (normal) and treated cells. Primers used in the study of FtsZ gene expression are listed in the S1–S12 Figs and S1–S4 Tables.

### 2.18. Statistical analysis

All the experiments were conducted in triplicates and values are depicted as Mean ± Standard Deviation. One-way ANOVA has been performed on required cases to verify the statistical significance between the treatments and Brown-Forsythe was performed as a post-hoc test for the experiment.

## 3. Results and discussion

Due to the resistance acquired by the various mycobacterial strains to the already existing drugs, development of novel and effective drug inhibiting newer targets has become an urgent need [50]. Medicinal plants and their constituents could possibly help in identifying new drugs to overcome the disease.

### 3.1. Characterisation of plant extracts by GCMS and HPLC

The presence of D-Pinitol in the three HXM plant extracts were confirmed by both GCMS and HPLC analysis.

**3.1.1. Gas Chromatography Mass Spectrometry (GCMS) analysis.** The various active compounds as identified by the database with their peak number, concentration (peak area %), and retention time (RT) of all three plant extracts and their biological activities of the phytochemicals identified by GCMS analysis are provided in the S1–S12 Figs and S1–S4 Tables. GCMS of three HXM plant extracts *A.nilotica*, *A.marmelos* and *G.glabra* shows the presence of D-Pinitol at the RT of 5.162, 5.160 and 5.169 respectively (Table 1). Mass fragmentation of *A.nilotica* at 5.162 min RT and *G.glabra* at 5.169 min RT showed 90% similarity index for D-Pinitol. *A.marmelos* at 5.160 min RT showed 92% of similarity index (S1–S12 Figs and S1–S4 Tables).

**3.1.2 High Performance Liquid Chromatography (HPLC).** The compound D-Pinitol is kept as a standard to internalize the comparison with all the three extracts. Standard D-Pinitol produced the optimized peak at 10.198 retention time. Similarly, extracts of *A. nilotica*, *A. marmelos* and *G. glabra* showed the presence of D-Pinitol at the retention time of 10.187, 10.143 and 10.135 respectively at the $10^{th}$ minute. They occupied the percentage area as 45.12, 48.33 and 55.02 (Table 1). This observation, clearly indicates the presence of natural compound D-Pinitol in the above three HXM extracts. So, D-Pinitol is considered as a compound of interest and taken for further study.

D-Pinitol commonly called as, 3-O-methyl D-Chiro inositol, is reported to have pharmacological significance, including antidiabetic, antiinflammatory, antioxidant and immunosuppressive potential [51]. It is used to treat AIDS, cardiovascular problems, hypertension, rheumatism and certain neurological problems [52]. D-Pinitol, a well known antidiabetic agent which acts by reducing the hyperglycemic levels [53].

## 3.2. Antimycobacterial activity of plant extracts

The antimycobacterial activity of the HXM extracts of *A. nilotica*, *A. marmelos* and *G. glabra* are 1.56 ± 0.03 mg/mL, 1.32 ± 0.02 mg/mL, 1.25 ± 0.03 mg/mL, respectively indicating that hexane methanol wash of the plants enhances the activity. D-Pinitol, a predominant phytochemical found in these plants (based on GCMS and HPLC profile) showed MIC of 0.11 ± 0.01 μg/mL, while commercial drugs namely INH and RIF had a MIC of 4.0 ± 0.06 μg/mL and 2.0 ± 0.04 μg/mL respectively (Table 2).

Others have also reported that the activity of different plant extracts depend on the solvent used for extraction. Hexane extract of *A. marmelos* fruit inhibited *M. tuberculosis* strain H37R$_V$ growth at 50 μg/mL while methanol extract inhibited at 100 μg/mL [54]. Acetone extract of *G. glabra* root inhibited *M. tuberculosis* strain H37R$_V$ in the range of 0.97–1.95 μg/mL [3]. Methanol followed by ethylacetate fraction of pomegranate peel inhibited *M.*

**Table 1. GCMS and HPLC analysis of D-Pinitol with the retention time and percentage area of the HXM plant extracts.**

| S.NO | Name | Retention time | % Area |
|---|---|---|---|
| | | **GCMS** | |
| 1 | *A.nilotica* | 5.162 | 31.83 |
| 2 | *A.marmelos* | 5.160 | 5.86 |
| 3 | *G.glabra* | 5.169 | 10.98 |
| | | **HPLC** | |
| 4 | Standard D-Pinitol | 10.198 | 100.00 |
| 5 | *A.nilotica* | 10.187 | 45.12 |
| 6 | *A.marmelos* | 10.143 | 48.33 |
| 7 | *G.glabra* | 10.135 | 55.02 |

**Table 2. MIC of HXM plant extracts.**

| Name of the plant / Compound | Common Name | MIC of HXM plant extracts |
|---|---|---|
| *Acacia nilotica* | Babool (Leaves) | 1.56 ± 0.03 mg/mL |
| *Aegle marmelos* | Bael/Vilvam (Unriped fruit pulp) | 1.32 ± 0.02 mg/mL |
| *Glycyrrhiza glabra* | Licorice (Dried Root) | 1.25 ± 0.03 mg/mL |
| D-Pinitol | - | 0.11 ± 0.01 µg/mL |
| Isoniazid | | 4.00 ±0.06 µg/mL |
| Rifampicin | | 2.00 ± 0.04 µg/mL |

*smegmatis* mc$^2$155 between 3.35 mg/mL to 0.395 mg/mL [12]. Glycyrrhizin, a phytochemical from *G. glabra* inhibited *M. tuberculosis* strain H37R$_V$ at 100 µg/mL [55]. The plants in this study seem to exhibit antimycobacterial activity.

## 3.3. Combinatorial interaction between plant extracts / D-Pinitol and commercial drugs like isoniazid and rifampicin

The extracts of *A. nilotica*, *A. marmelos*, *G. glabra* and D-Pinitol are studied in conjunction with INH and RIF in different proportions to test the effect of combination therapy against *M. smegmatis*. The results indicated that, these at sub-minimal inhibitory concentration were able to reduce the MIC of both the drugs. HXM extract of *A. nilotica* (1.56 mg/mL) reduced the MIC of INH from 4.00 µg/ mL to 0.01 µg/ mL and MIC of RIF from 2.00 µg/ mL to 0.01 µg/ mL (FIC index = 0.29 and 0.31 respectively). HXM extract of *A. marmelos* (1.32 mg/ml) reduced the MIC of INH from 4.00 µg/ mL to 0.05 µg/ mL and that of RIF from 2.00 µg/ mL to 0.01 µg/ mL (FIC index = 0.27 and 0.29 respectively). HXM extract of *G. glabra* (1.25 mg/mL) decreased the MIC of INH from 4.00 µg/ mL to 0.02 µg/ mL and that of RIF from 2.00 µg/ mL to 0.01 µg/ mL (FIC index = 0.18 and 0.17 respectively). D-Pinitol (0.11 µg/ mL) reduced the MIC of INH from 4.00 µg/ mL to 0.01 µg/ mL and that of RIF from 2.00 µg/ mL to 0.01 µg/ mL (FIC index = 0.16 and 0.18) respectively. These results indicate that these three HXM plant extracts and D-Pinitol combine in synergistic fashion with these two drugs their by reducing their concentration by (8–10 folds) required to inhibit *M. smegmatis*.

Published literature indicated that, MIC of RIF is reduced by 64-fold in combination with the acetone extract of *Cremaspora triflora* Thonn against *M. smegmatis* ATCC 1441 [56]. MIC of INH is reduced by 8 fold in conjunction with ethyl acetate extract of *Knowltonia vesicatoria* L against *M. smegmatis* mc2 155 [57]. Plumbagin, from *Plumbago zeylanica* L and Ferelenol from *Ferelenol communis* reduced the MIC of INH by 8 fold against *M.tuberculosis* strain H37R$_V$ [58]. Also, Curcumin and demethoxycurcumin reduced the MIC of INH by 16 fold against *M. smegmatis* mc$^2$ 155 [59]. MIC of RIF reduced from 0.015 to 0.00048 mg/mL when given in combination with antitubercular drug SQ109 against *M. tuberculosis* strain H37R$_V$ [60]. Cumene hydroperoxide reduced the MIC of RIF by 16-fold against *M. tuberculosis* strain H37R$_V$ [61].

Though D-Pinitol has been considered as the lead component to test for activity since being a major fraction in GCMS and HPLC analysis, it can also be noted that in addition to D-Pinitol other phytochemicals present in the HXM extracts may also contribute to the anti- mycobacterial activity which may require separate study. Retention Time (RT) comparison with the standard (D-Pinitol) was the only method used in this study to prove the presence of D-Pinitol in the plant extracts. The results of the combination study shown in the form of isobologram (Figs 1 and 2) clearly indicates the potential of the HXM extracts as supplement to the drugs at lower concentration for inhibition of *M.smegmatis*.

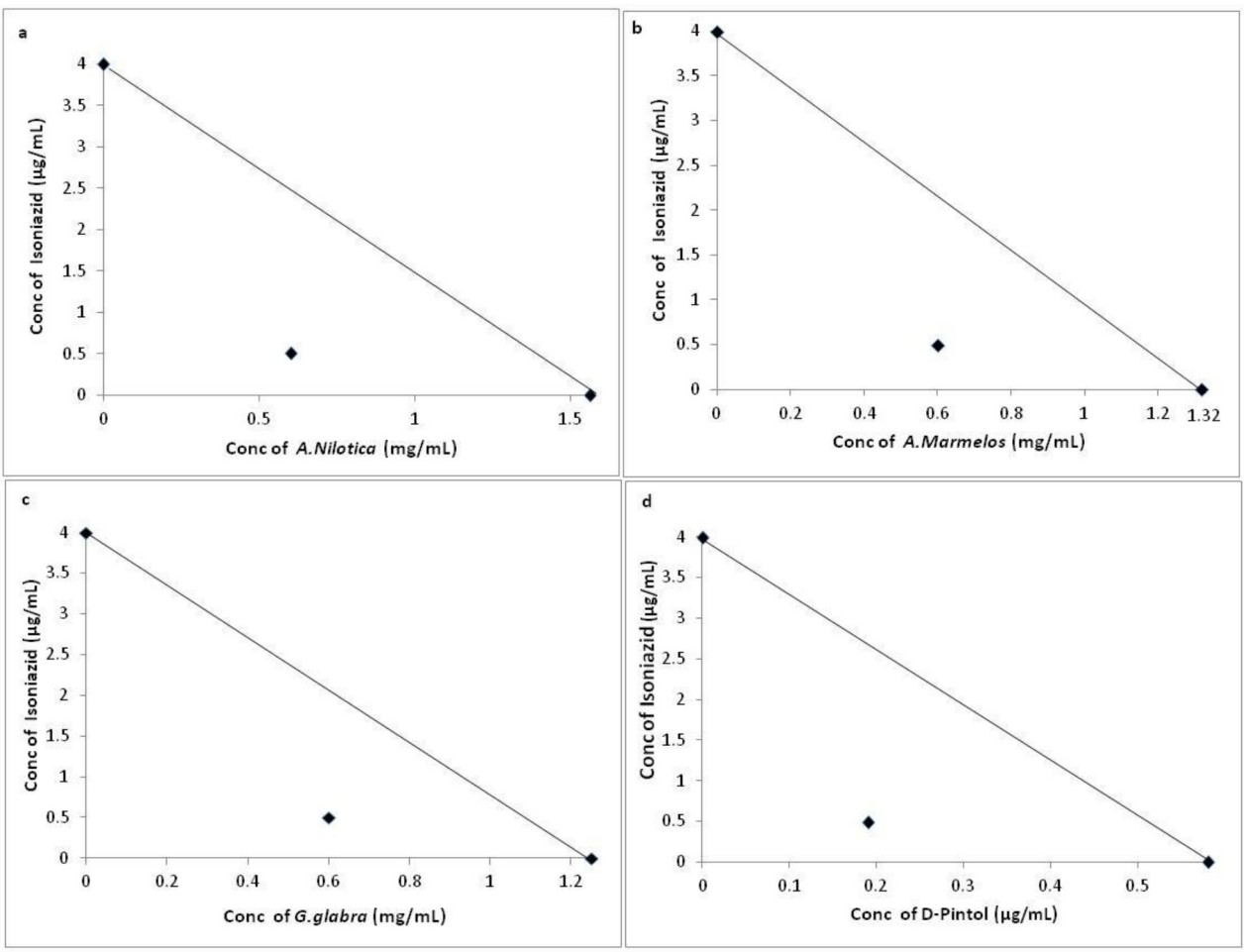

**Fig 1. Interaction of Isoniazid with HXM extracts of a)** *A. nilotica*, **b)** *A. marmelos* **and c)** *G. glabra* **and d) the phytochemical, D-Pinitol against** *M. smegmatis.* Line indicates additivity. Point below the line indicates synergistic interaction between the two drugs. FIC Index a) 0.29, b) 0.27, c) 0.18, d) 0.16.

### 3.4. Time–kill studies

Extracts of *A. nilotica*, *A. marmelos* and *G. glabra* and D-Pinitol were employed for time-kill studies individually and in combinations with INH and RIF. The values of CFU at 24th hour were represented in Table 3. *M smegmatis* doubling time is 6–8 hours [62]. At 24th hour, drug or extracts are not able to prevent the growth of *M.smegmatis*. An average of 3 Log fold decline in the growth rate of *M. smegmatis* post 48 hours when compared to the untreated control was observed in all the cases. Also the extracts of these plants in combination with INH and RIF showed an average 2 Log reduction in viable cell count when compared to untreated while other *A.nilotica* extract in combination showed an equivalent kill-profile similar to INH and RIF alone. The results are given in S1–S12 Figs and S1–S4 Tables. To confirm whether the growth rate retardation activity is governed by time or due to the presence of active constituents of individual extracts, the activity of D-Pinitol in combination, was also investigated which also showed that in combination with INH (Fig 3a) and RIF (Fig 3b) a 2 Log retardation in the growth of viable cell count when compared to control was observed. Literature reports that, dihydrofusarubin and RIF combination showed bacteriostatic activity [63]. Ethanolic

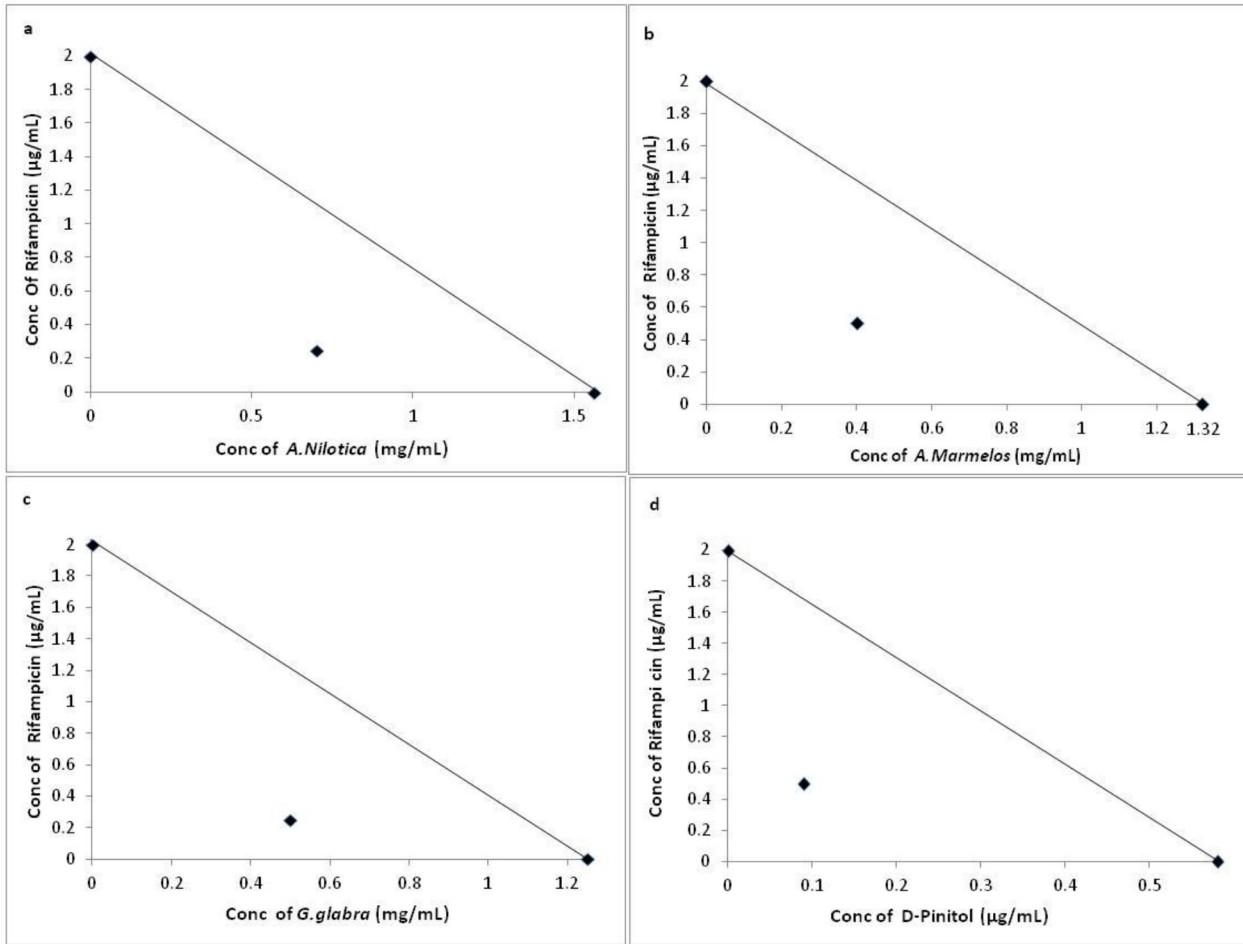

**Fig 2. Interaction of rifampicin with extracts of a)** *A. nilotica*, **b)** *A. marmelos* **and c)** *G. glabra*, **and d) the phytochemical, D-Pinitol against** *M. smegmatis*. Line indicates additivity. Points below the line indicates synergistic interaction between the two. FIC Index a) 0.31, b) 0.29, c) 0.17, d) 0.18.

extract of *Andrographis paniculata* Burm showed bacteriostatic activity against *M. tuberculosis* in combination with INH [64].

Minimum inhibitory concentration (MIC) is defined as the lowest drug concentration that shows 90% growth inhibition of *M. smegmatis* [65], [66]. At 1 x MIC, 10% of the organisms which survive at this specific concentration will acclimatize to this conditions and replicates at further time intervals. Thus, an increase in the CFU of the organism is observed in the graph as a function of time (Fig 3a and 3b). In our study, combination of the drugs and HXM extracts have much less reduction when compared to the extracts or D-Pinitol alone. This decrease could be due to formation (Millard reaction) of adduct between INH/RIF through the amine group with D-Pinitol (sugar). This Millard reaction between reducing sugars and amines were reported in studies[67].

## 3.5. Live/ Dead assay

Membrane damage of *M. smegmatis* cell wall after incubating with HXM extracts and D-Pinitol were determined using a Fluorescent Microscope, after staining with STY09 and PI (Fig 4a–4g). Green indicates live and red indicates dead cells, since PI penetrates the damaged cell

**Table 3. CFU/mL for control and treated samples observed at 24 hrs.**

| Treatment | CFU/mL at $10^{-2}$ dilution |
|---|---|
| Control | 614± 3.35 |
| INH | 522± 4.87 |
| RIF | 502± 3.51 |
| *A.nilotica* | 456±1.89 |
| *A.nilotica*/ INH | 416±2.32 |
| *A.nilotica*/ RIF | 401±3.21 |
| *A.marmelos* | 445±1.91 |
| *A.marmelos*/ INH | 404±1.82 |
| *A.marmelos*/RIF | 411±2.67 |
| *G.glabra* | 434±3.76 |
| *G.glabra* /INH | 391± 4.67 |
| *G.glabra* /RIF | 386±4.56 |
| D-Pinitol | 401±1.96 |
| D-Pinitol/ INH | 294±1.75 |
| D-Pinitol/RIF | 302±2.25 |

wall. Treatment with *A. marmelos*, *A. nilotica*, *G. glabra* and D-Pinitol leads to 43.4%, 55.1%, 49.1% and 50.4% cell wall damage respectively (Fig 4h). Percentage of cell death on treatment with INH and RIF are 31.7% and 49.3% respectively. The observed cell death in treated bacteria which implies that chosen plant HXM extracts and D-Pinitol has the ability to rupture the cell membrane by releasing intracellular contents and cause death.

Membrane structural integrity of bacterial cell plays a vital factor for the function of the cell membrane by internally maintaining cellular conditions required for metabolism [68]. Numerous studies have reported that, plant phytoconstituents play a role in rupturing cell membrane by affecting its structural integrity. Curcumin I was reported to be active against both Gram positive and negative bacteria, with 100% killing at a dose of 36 μg/mL [69]. Present study indicates that the concentration of D-Pinitol at 0.1l μg/mL shows cell membrane damage.

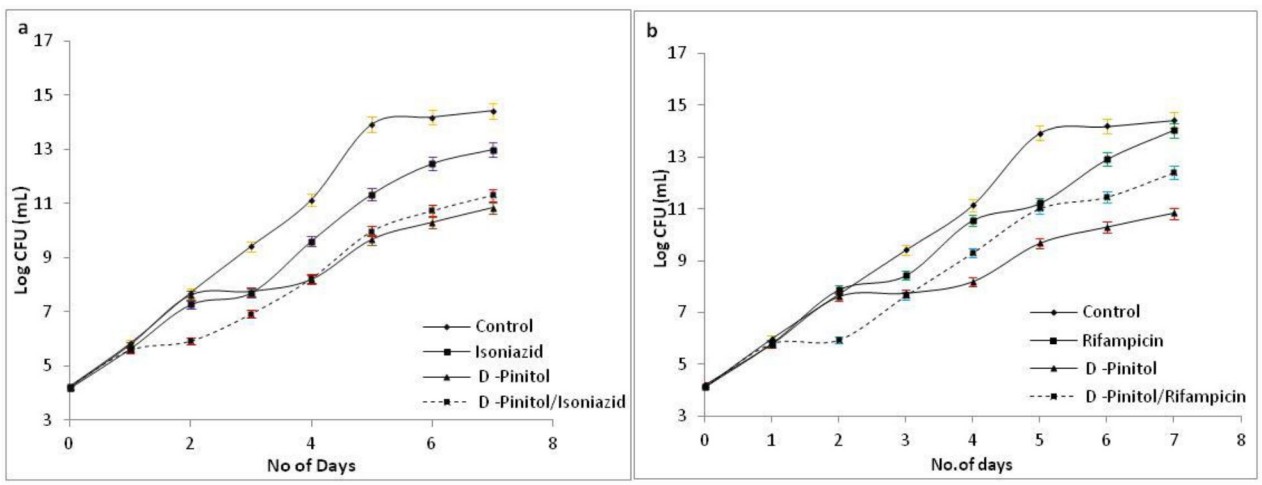

**Fig 3. Time kill curves of a) isoniazid, b) rifampicin in combination with the phytochemical D-Pinitol against *M. smegmatis*.**

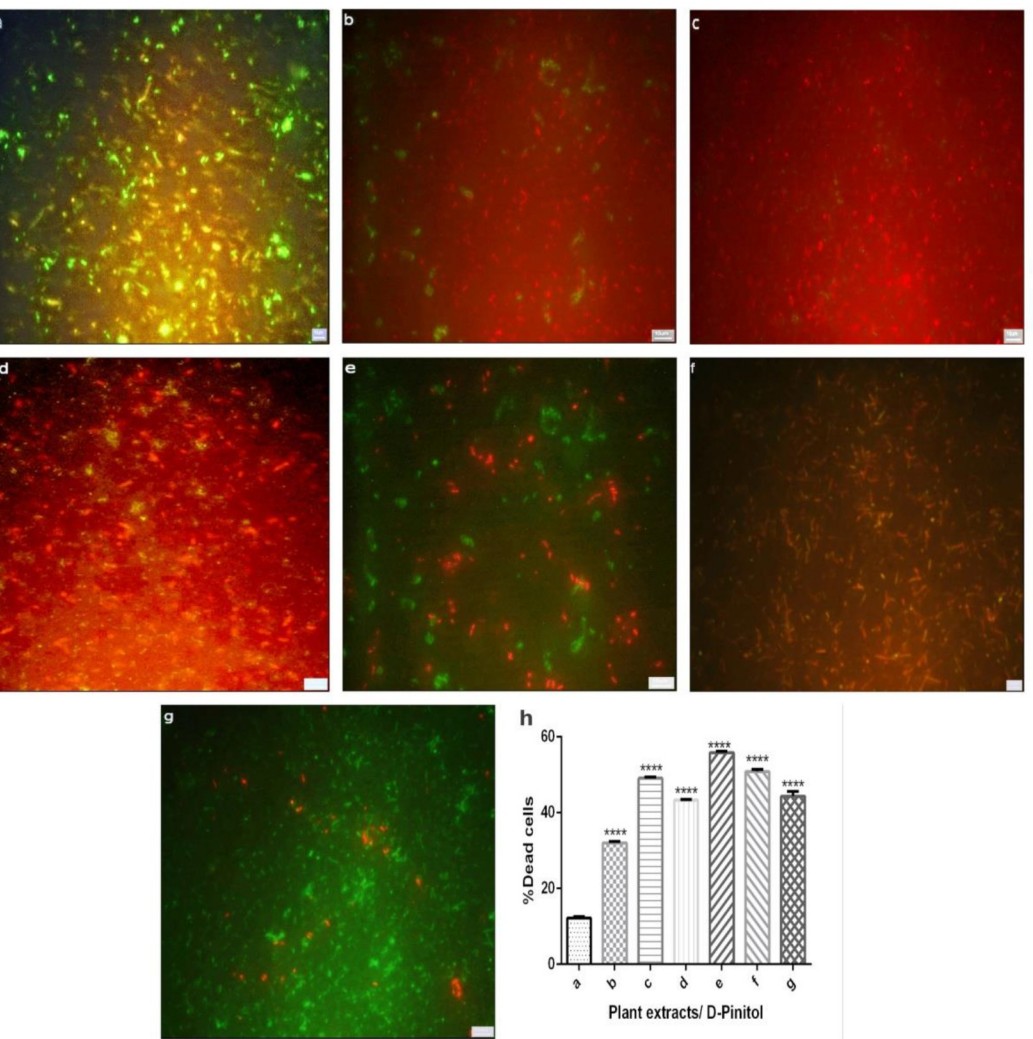

**Fig 4. Fluoroscence images of *M. smegmatis* cells on treatment with a) control, b) isoniazid, c) rifampicin, d) *A. marmelos*, e) *A. nilotica*, f) *G. glabra* and g) D-Pinitol [Green indicate viable and Red indicates dead (membrane damaged) cells] h) % of dead cells on treatment (estimated using Baclight dual staining kit).** P = **** <(0.0001) when compared with the control.

## 3.6. Cell elongation study

Microscopic images of *M. smegmatis* on exposure to the plant extracts, the phytochemical D-Pinitol and drugs are shown in Fig 5a–5g. The length of *M. smegmatis* (3.9 ± 0.5 μm) on treatment with HXM extract of *A. nilotica* (8.2 ± 1.0 μm), *A. marmelos* (15.7 ± 1.2 μm), *G. glabra* (13.5 ± 0.5 μm) and D-Pinitol (12.7 ± 0.1 μm) increased by three to five times. INH and RIF treated *M. smegmatis* cells also exhibited elongation (9.7 ± 0.6 μm and 11.5 ± 0.2 μm). Histogram of distribution of cell lengths of the control and treated *M.smegmatis* bacteria (Fig 5h) created using ImageJ clearly shows the result of treatment of different combinations.

Plumbagin and totarol increased the bacteria length by seven and five folds [70]. Length of *M. smegmatis* bacteria increased to three to four folds on treatment with Esculetin and Scopuletin [12]. We also observed similar results with these HXM extracts and D-Pinitol indicating that they probably act by inhibiting the first stage of cell division proteins, namely FtsE, FtsX

(a)

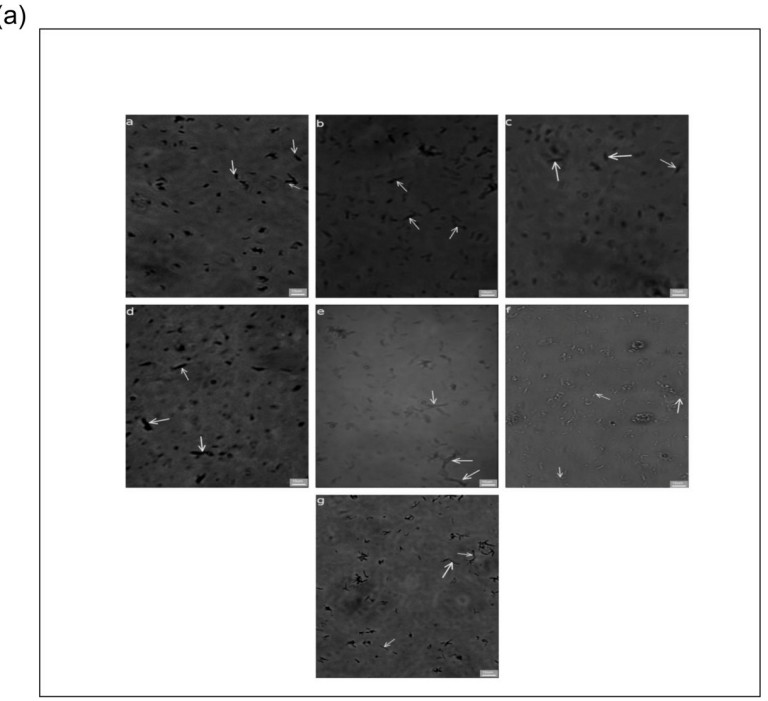

(b)

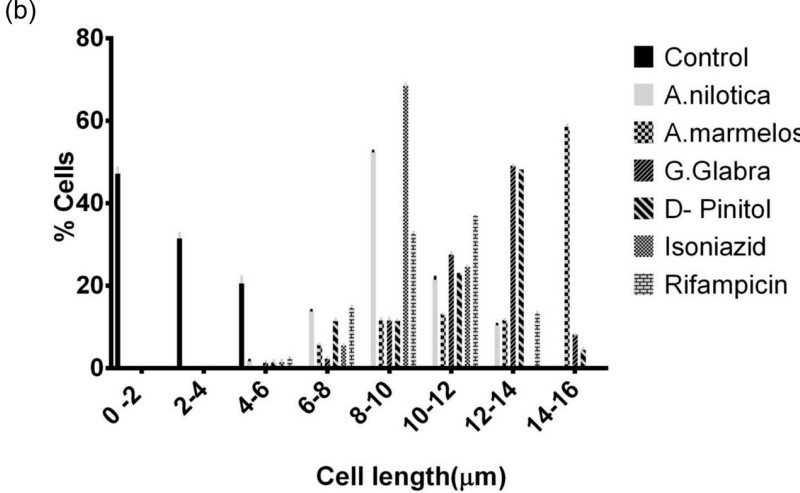

**Fig 5.** (**a**) Elongation of *M. smegmatis* cells on treatment with a) control, b) isoniazid, c) rifampicin, d) *A. marmelos*, e) *A. nilotica*, f) *G. glabra* and g) D-Pinitol, arrow indicates elongated cells. h. (**b**) Represents the histogram of distribution of cell lengths of the control and treated *M.smegmatis* bacteria as measured using Image J. Three different fields were observed for each of the microslide and 100 bacterial cells were counted per field.

and FtsZ by inducing cell elongation thereby causing cell death. The elongated cells could not undergo cell division this could be due to the nonfunctional septum formation by their action.

### 3.7. Effect of plant extracts on *M. smegmatis* FtsZ GTPase activity

Since the plant extracts were shown to interfere with cellular morphology leading to elongated cells. The effect of these plant extracts on cell division were probed. Since FtsZ is the prime

protein that acts as a choreographer to organize and assemble other cell division machinery, we probed the effect of HXM extracts of these plants over FtsZ activity.

*A. nilotica*, *A. marmelos* and *G. glabra* inhibited the *M.tb* FtsZ GTPase activity with an IC$_{50}$ of 1.399, 1.329 and 1.564 mg/mL respectively (Fig 6a and 6b). D-Pinitol inhibited GTPase activity with an IC$_{50}$ of 1.524 µg/mL. Berberine, served as a positive control here and it inhibited the activity of GTPase with an IC$_{50}$ of 1.861 µg/mL. Compounds like 7-dimethyl-4-methylcoumarin, daphnetin and esculetin have also been reported to inhibit this activity [12]. GTPase activity of FtsZ protein is essential for the creation of Z-ring and abnormalities in this function caused by plant based inhibitors will lead to cell death.

### 3.8. Polymerisation of FtsZ

In order to further confirm the effect of extracts on polymerisation of the FtsZ, a light scattering experiment is performed using HXM extracts / D-Pinitol at 1.5 x MIC. All the three plant extracts showed a decrease in the rate of protofilament by 50% after 300 seconds, whereas D-Pinitol showed a stable reduction (Fig 6c). Many natural compounds from plant extracts such as Cinnamic acid, Curcumin [71] and Berberine [72] are reported to inhibit the polymerisation activity.

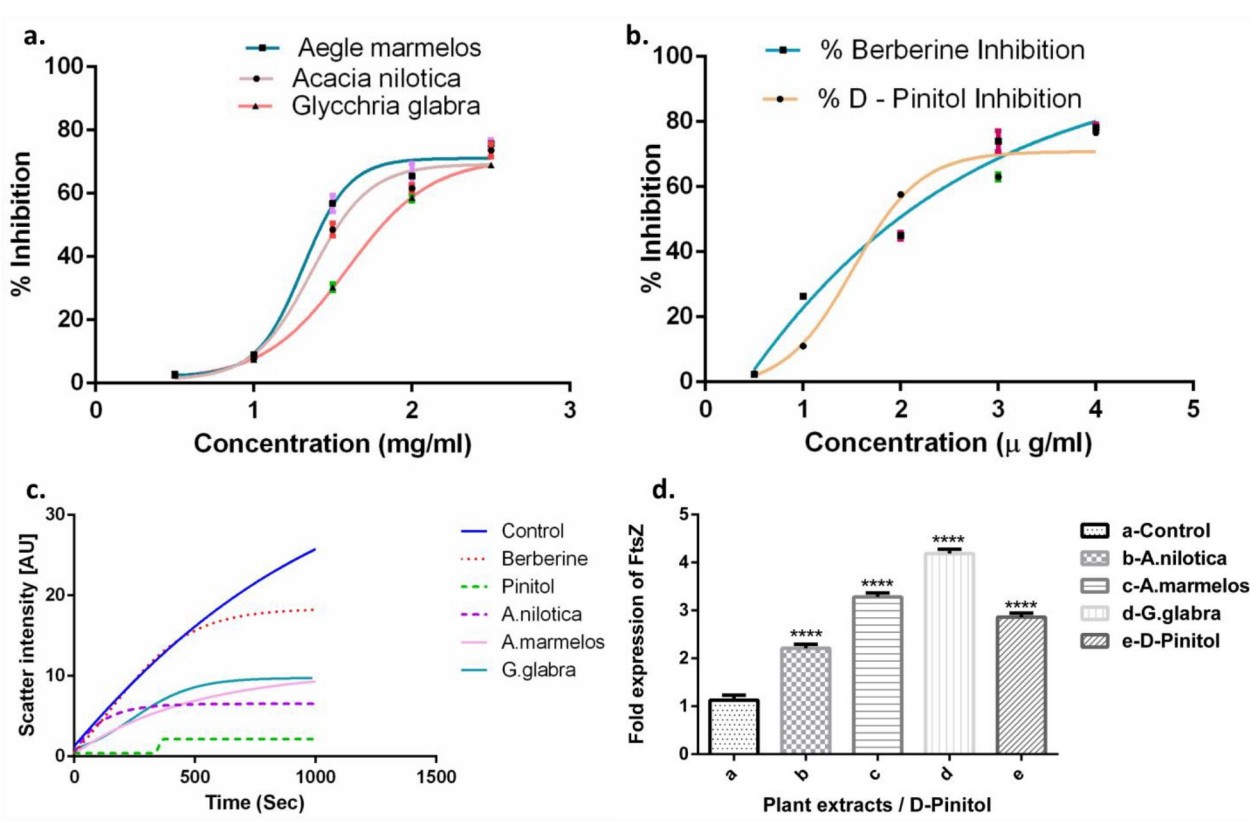

**Fig 6.** Effect of a) HXM extracts (mg/mL) and b) D-Pinitol (µM) on *M. smegmatis* FtsZ - GTPase polymerisation activity. c) FtsZ polymerisation activity measured using 90° light scattering assay at 1.5 x MIC with FtsZ at 12 µM. Each point represents the mean of three different replicates, d) RT–PCR Gene expression analysis—fold increase in the expression of FtsZ post treatment at 1.5 x MIC. P = **** <(0.0001) when compared with the control.

### 3.9. Studies on gene expression of FtsZ

In our studies, it was observed that the bacterial cell membrane damage and cell elongation of the treated *M.smegmatis* cells take place after 4 hr. Based on this, the gene expression studies of FtsZ was performed after 4 hr incubation. The concentration of HXM extracts and D-Pinitol increased to 1.5 x MIC to exhibit a fold increase in expression of FtsZ gene. The FtsZ gene expression increased by 2.15, 3.34 and 4.19 fold on treatment with *A. nilotica*, *A. marmelos* and *G. glabr*a when compared to untreated. D-Pinitol treatment increased the FtsZ gene expression by about 2.80 fold (Fig 6d). This indicates that these HXM extracts / D-Pinitol alters the gene expression of FtsZ considerably in addition to affecting the activity of the FtsZ protein. Many natural phenolic compounds and synthetic derivatives of coumarins are reported for the increased expression of *M. tuberculosis* FtsZ [12].

## 4. Conclusions

*M. smegmatis*, a non pathogenic bacteria, which is closer to *M. tuberculosis* in sequence similarity is used in our study to find the effect of plant extracts against it. The *in vitro* synergistic activity of extracts of *A. nilotica*, *A. marmelos* and *G. glabr*a and the phytochemical D-Pinitol (which is found in all three plants extracts) in combination with two drugs namely INH and RIF were also studied. Plant extracts are known to act on multiple targets while INH is known to inhibit the mycolic acid production, a cell wall component [73] and RIF is known to act by inhibiting the bacterial DNA dependent RNA synthesis by inhibiting the RNA [74].

HXM extracts and the phytochemical, D-Pinitol produced synergistic effects at different subminimal concentration when combined with the drugs. In time kill analysis, treatment of log-phase *M. smegmatis* with 1 x MIC concentration resulted in 2 to 3 Log reduction in viable cells indicating that HXM extracts exhibit bacteriostatic activity. The observed decrease in the activity when HXM plant extracts/ D- Pinitol and drugs are combined together could be due to formation (Millard reaction) of adduct between INH/RIF through the amine group with D-Pinitol (sugar). HXM extracts and D-Pinitol also induce damage to the cell membrane of *M. smegmatis* by affecting its structural integrity thereby causing cell death.

D-Pinitol is an interesting biological niche that inhibits *M. smegmatis* growth, probably by perturbing the cell division process which resulted in an elongated morphology. This implies that it could be a potential lead as an antimycobacterial agent in drug development. So, for the first time the current study has identified a new phytochemical, which could lead a way towards the development of novel antimycobacterial drug. Also, synergy study has indicated that concentration of commercial drug such as INH and RIF could be reduced considerably by combining them with HXM plant extracts which could lead to reduced toxicity of the former and address the emergence of resistant strains.

The current study indicates that the chosen plant extracts / D-Pinitol can be explored for potential leads in tuberculosis therapy. The study has also identified that they affect the cell division process by inhibiting the GTPase activity of Mycobacterium-FtsZ. These plant extracts also altered the expression of *Mycobacterium*-FtsZ gene. Of course, the pharmacokinetics and pharmcodynamics of these combinations need to be studied in animal models before these could be actually taken up further as adjuvant therapy to potentiate the action of commercial drugs or as complementary and alternative therapy.

## Supporting information

**S1 Table. GTPase IC$_{50}$ of the HXM plant extracts.**
(DOCX)

**S2 Table. Effect of HXM extracts / D-Pinitol on cell elongation of *M. smegmatis*.**
(DOCX)

**S3 Table. List of primers used in the FtsZ gene expression study.**
(DOCX)

**S4 Table. Relative expression levels of FtsZ.**
(DOCX)

**S1 Fig. GCMS peaks and table values *of A.nilotica*.**
(DOCX)

**S2 Fig. GCMS peaks and table values *of A.marmelos*.**
(DOCX)

**S3 Fig. GCMS peaks and table values *of G.glabra*.**
(DOCX)

**S4 Fig. Mass fragmentation of *A.nilotica*.**
(DOCX)

**S5 Fig. Mass fragmentation of *A.marmelos*.**
(DOCX)

**S6 Fig. Mass fragmentation of *G.glabra*.**
(DOCX)

**S7 Fig. HPLC of *A. nilotica*.**
(DOCX)

**S8 Fig. HPLC of *A. marmelos*.**
(DOCX)

**S9 Fig. HPLC of *G. glabra*.**
(DOCX)

**S10 Fig. HPLC of D- Pinitol (standard).**
(DOCX)

**S11 Fig. Time kill curves of Isoniazid in combination with HXM extracts of a) *A. nilotica*, b) *A. marmelos*, and c) *G. glabra* against *M. smegmatis*.**
(DOCX)

**S12 Fig. Time kill curves of rifampicin in combination with HXM extracts of a) *A. nilotica*, b) *A. marmelos*, and c) *G. glabra* against *M. smegmatis*.**
(DOCX)

## Acknowledgments

We also thank Prof. Anju Chada and Prof. Guhan Jayaraman, Department of Biotechnology, IIT Madras for analyzing our plant extracts for GCMS and HPLC.

## Author Contributions

**Investigation:** Radhika Ravindran.

**Methodology:** Gayathri Chakrapani.

**Supervision:** Mukesh Doble.

**Visualization:** Kartik Mitra.

**Writing – original draft:** Radhika Ravindran.

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
