## [Decision Letter · Decision Letter 0]

20 Nov 2019

PONE-D-19-29098

Antitubercular activity of plant extracts and their action on cell division protein (FtsZ)

PLOS ONE

Dear Dr Ravindran,

Thank you for submitting your manuscript to PLOS ONE. After careful consideration, we feel that it has merit but does not fully meet PLOS ONE’s publication criteria as it currently stands. Therefore, we invite you to submit a revised version of the manuscript that addresses the points raised during the review process.

ACADEMIC EDITOR: Please Pay close attention to the reviewer comments and incorporate all clarification on the manuscript in addition to point-by-point response.

We would appreciate receiving your revised manuscript by Jan 04 2020 11:59PM. To enhance the reproducibility of your results, we recommend that if applicable you deposit your laboratory protocols in protocols.io, where a protocol can be assigned its own identifier (DOI) such that it can be cited independently in the future. For instructions see: http://journals.plos.org/plosone/s/submission-guidelines#loc-laboratory-protocols

We look forward to receiving your revised manuscript.

Kind regards,

Selvakumar Subbian, Ph.D.

Academic Editor

PLOS ONE

Journal Requirements:

1, In your Data Availability statement, you have not specified where the minimal data set underlying the results described in your manuscript can be found. PLOS defines a study's minimal data set as the underlying data used to reach the conclusions drawn in the manuscript and any additional data required to replicate the reported study findings in their entirety. All PLOS journals require that the minimal data set be made fully available. For more information about our data policy, please see http://journals.plos.org/plosone/s/data-availability.

Reviewers' comments:

Reviewer's Responses to Questions

**Comments to the Author**

1. Is the manuscript technically sound, and do the data support the conclusions?

Reviewer #1: Partly

Reviewer #2: Yes

Reviewer #3: No

2. Has the statistical analysis been performed appropriately and rigorously? 

Reviewer #1: No

Reviewer #2: Yes

Reviewer #3: No

3. Have the authors made all data underlying the findings in their manuscript fully available?

Reviewer #1: Yes

Reviewer #2: Yes

Reviewer #3: No

4. Is the manuscript presented in an intelligible fashion and written in standard English?

Reviewer #1: No

Reviewer #2: No

Reviewer #3: Yes

5. Review Comments to the Author

Reviewer #1: Manuscript Number: PONE-D-19-29098

Manuscript Title: Antitubercular activity of plant extracts and their action on cell division protein (FtsZ)

The article indicated above has some major revisions that needs to be addressed and inappropriate to be accepted in the present version. Many claims have been made that needs further clarification before it can be published in your esteemed journal. The writing and formatting needs some more improvisation too.

I have listed my concerns which needs to be addressed before being accepted for publication.

The article titled “Antitubercular activity of plant extracts and their action on cell division protein (FtsZ)” talks about possible synergistic effects of plant extracts with anti-mycobacterial drugs which is tested using fast grower M. smegmatis. The title is quite misleading to any reader with the use of word “anti-tubercular activity”. It is preferred to avoid using the term “anti-tubercular activity” in the title.

In line 192, under REMA assay under materials and methods section, Rif and Inh is used at 1 mg/ml. This contradicts with several earlier reports for MIC values of Rifampicin and isoniazid which are different as estimated visually or by alamar blue or resazurin assay.

Line 193 under same assay says “The plate was wrapped using aluminium foil and incubated overnight at 37°C for 48 hour”. Kindly clarify if left overnight or for 48 hours. This could be useful to anyone who wants to repeat the assay. Besides, drug treatment for MIC determination would be done more than 24 hours for M. smegmatis.

Line 195-196 under same assay says “The change in colour of the solution from blue to pink indicates inhibition while no change indicates the growth of bacteria”. Actually, the resazurin colour change to pink indicates growth and the statement gives the impression the author(s) is unfamiliar with the protocol.

In line 205, authors refer to synergy drug testing for 24 hours at 37’C and report differential FIC index. However, in Fig 3 detailing time kill analysis we see no difference in CFU as indicated in the graph. What could be the possible explanation for this. Besides, it would be appropriate to include the error bars with the mean values plotted in graph.

In line 214, time kill assay was performed at 1x MIC of the drug. However, routinely synergistic drug effects would be tested at sub MIC levels.

In line 271, to study FtsZ gene expression, plant extracts and drugs have been tested at 1.5 x MIC. Is there any reason to use such higher value? Clarity on this will help understand why it was used.

In line 282, the authors have written that “There has been no new anti tuberculosis drug introduced in the past 30 years”. Please remove this statement. The authors could have been more careful while writing this.

The Food and Drug Administration (FDA), on 28 December 2012, granted accelerated approval to Janssen’s SIRTURO™ (bedaquiline) tablets as a part of combination therapy in adults with multi-drug-resistant TB (MDR-TB). It is the first new anti-TB drug to be approved after 1998 (rifapentine was approved in 1998) and the first anti-TB drug with a novel mechanism of action to be approved after 40 years (rifampicin was approved in 1974 (Deoghare S. Bedaquiline: a new drug approved for treatment of multidrug-resistant tuberculosis. Indian J Pharmacol. 2013;45(5):536–537). Besides, Delamanid is approved by the EMA and is marketed under the trade name Deltyba as oral tablets. It is marketed by Otsuka Pharmaceutical Co., Ltd, Tokyo, Japan (Skripconoka V, Danilovits M, Pehme L, Tomson T, Skenders G, Kummik T, Cirule A, Leimane V, Kurve A, Levina K, Geiter LJ, Manissero D, Wells CD: Delamanid improves outcomes and reduces mortality in multidrug-resistant tuberculosis. Eur Respir J. 2013 Jun;41(6):1393-400).

In line 324, the authors claim that “The results indicated that, these at sub-minimal inhibitory concentration were able to reduce the MIC of both the drugs”. However, the concentration of the HXM extracts used here are given as the same the MIC values given in supplementary table S4. The statement needs clarification.

In lines 363-365, the authors claim 2-log growth retardation when D-Pinitol was used in combination with Rifampicin or Isoniazid. However, in figure 3, D-pinitol by itself shows better/similar retardation independent of anti-TB drugs. What is the possible explanation of this ??

Reviewer #2: The manuscript is not written in good English. There are many grammatical, topological and convention related errors some of which are mentioned below.

Comments

Minor:

1. Title is incomplete. Title must include the organism worked upon.

2. Abstract: Do not include the objective as first line of abstract. It should not start with “To…..”

3. Abbreviate the name of the organism/plant wherever its used for the first time in the text.

4. Units used in the text are not constant. Some values are represented in mg/ml, ug/ml and some in uM. It is very difficult for the readers to understand and compare.

5. Line 22: Hexane and not ‘hexame’

6. Line 39: in vitro should be italicized here as well as throughout the text.

7. Line 59: H37RV is incorrect throughout the text. Correct as H37Rv.

8. Section 2.1 says 7H9 broth was procured from Sigma and section 2.5 says the same is from BD. Please correct.

9. Line 269: Correct the gene name as 16S rRNA

10. Line 269: Write Mycobacterium sp. instead of only genus.

11. Line 283: ‘strain of existing drug’ does not make sense

12. Line 297: Write observation instead of statement

13. Line 323: Remove the author’s name for reference 51. Just use number.

Major:

14. Section 2.10:

a) If you added plant extract samples after adding 100ul broth, it will change the volume. How did you keep the volume constant?

b) Why did you use 1mg/ml INH and RIF and not test other concentrations?

c) Did you use medium control without DMSO as a control for INH?

d) All volumes used should be fixed. Do not write about 20ul and so.

15. Why did you use 1X MIC for time kill assay and 1.5X for gene expression studies. Give reference or explanation for selecting these numbers.

16. Section 3.2, Line 307: ‘hexane wash of plants enhances the activity’..did you study unwashed extracts to compare.

17. Section 3.3: Paragraph 1 results can be represented as a small table for an easy comparison.

18. Conclusion: Include references for mode of action of INH and RIF (line 442-444).

Reviewer #3: The phytochemistry that was done to determine the presence and quantity of pinitol is not clear. The paper is lacking any statistical analysis, although this is mentioned in the M&M section.

Please find further comments in the attached PDF.

6. PLOS authors have the option to publish the peer review history of their article (what does this mean?). If published, this will include your full peer review and any attached files.

Reviewer #1: Yes: Dr. Radha Gopalaswamy

Reviewer #2: No

Reviewer #3: No

---

## [Author Response · Author response to Decision Letter 0]

10 Jan 2020

Reviewer #1: Manuscript Number: PONE-D-19-29098

1, Manuscript Title: Antitubercular activity of plant extracts and their action on cell division protein (FtsZ).The article titled “Antitubercular activity of plant extracts and their action on cell division protein (FtsZ)” talks about possible synergistic effects of plant extracts with anti-mycobacterial drugs which is tested using fast grower M. smegmatis.

 The title is quite misleading to any reader with the use of word “anti-tubercular activity”. It is preferred to avoid using the term “anti-tubercular activity” in the title.

Response: We have changed the title to“ Inhibitory activity of traditional plants against Mycobacterium smegmatis and their action on Filamenting temperature sensitive mutant Z (FtsZ) - a cell division protein”. 

2, In line 192, under REMA assay under materials and methods section, RIF and INH is used at 1 mg/ml. This contradicts with several earlier reports for MIC values of Rifampicin and isoniazid which are different as estimated visually or by alamar blue or resazurin assay.

Response: The stock of RIF and INH is 1mg/mL in DMSO/ water, from which further 10X dilution was made to avoid DMSO effect. 100 µl from the stock concentration was made up to 1mL with the corresponding DMSO/ water. Such that maximum concentration of 100 µg/mL was used for the assay. We have corrected the above statement under material sections 

3, Line 193 under same assay says “The plate was wrapped using aluminium foil and incubated overnight at 37°C for 48 hour”. Kindly clarify if left overnight or for 48 hours. This could be useful to anyone who wants to repeat the assay. Besides, drug treatment for MIC determination would be done more than 24 hours for M. smegmatis.

Response: It should be written as, the plate was wrapped using aluminium foil and incubated at 37 °C for 48 hour. The term overnight was wrongly placed. We have corrected the above statement under material sections.

4, Line 195-196 under same assay says “The change in colour of the solution from blue to pink indicates inhibition while no change indicates the growth of bacteria”. Actually, the resazurin colour change to pink indicates growth and the statement gives the impression the author(s) is unfamiliar with the protocol.

Response: It should be written as the change in color from blue to pink indicates the growth of bacteria and no change indicates inhibition. The above changes were also made in the text file.

5, In line 205, authors refer to synergy drug testing for 24 hours at 37’C and report differential FIC index. However, in Fig 3 detailing time kill analysis we see no difference in CFU as indicated in the graph. What could be the possible explanation for this.

Response: In antibacterial study, the MIC was observed post 48 hr. hence there was no significant difference in Time kill at the first 24 hr, however the killing effect (Fig 3) can be seen after 48 hr.

6, In line 214, time kill assay was performed at 1x MIC of the drug. However, routinely synergistic drug effects would be tested at sub MIC levels.

Response: Yes, it should be tested under sub MIC levels. But, in synergistic studies, plant extracts acted in subminimal concentration along with the combination of INH/ RIF by reducing the MIC of first line drugs from 4.00 to 0.01 μg/ mL for INH and 2.00 to 0.01 μg/ m for RIF. So,synergy study was performed at lower concentration.

7, In line 271, to study FtsZ gene expression, plant extracts and drugs have been tested at 1.5 x MIC. Is there any reason to use such higher value? Clarity on this will help understand why it was used.

Response: The compounds showed antimycobacterial activity against M.smegmatis at 1x MIC at 48 hr incubation, but in the case of gene expression studies which is done in 4 hr, the effect of 1x MIC may not show significant difference. Hence in order to mitigate the shorter experiment time, we considered a step higher concentration of 1.5x MIC. 

8, In line 282, the authors have written that “There has been no new anti tuberculosis drug introduced in the past 30 years”. Please remove this statement. The authors could have been more careful while writing this.

Response: We removed the statement. Thank you for sharing the information on drugs.

9, In line 324, the authors claim that “The results indicated that, these at sub-minimal inhibitory concentration were able to reduce the MIC of both the drugs”. However, the concentration of the HXM extracts used here are given as the same the MIC values given in supplementary table S4. The statement needs clarification.

Response: In the study, it is observed that combination of the plant extract with current drugs (RIF/INH) were able to reduce the drug dosages required. Rifampicin and Isoniazid alone exhibited MIC at 2.0 µg/mL and 4.0 µg/mL , whereas in combination with the reported plant extracts they exhibited anti-tubercular activity at 0.01 µg/mL and 0.02 µg/mL. Thus we say that these plant extracts were able to synergize with RIF & INH to exhibit activity at their sub-minimal concentration.

10, In lines 363-365, the authors claim 2-log growth retardation when D-Pinitol was used in combination with Rifampicin or Isoniazid. However, in figure 3, D-pinitol by itself shows better/similar retardation independent of anti-TB drugs. What is the possible explanation of this ??

Response: At a concentration of 0.11 µg/mL D- Pinitol showed better MIC activity, whereas first line drug such as Rifampicin, Isoniazid showed MIC of 2.00 and 4.00 µg/ml. This may be the possible reason for D - Pinitol to show its enhanced activity against Mycobacterium smegmatis.

Reviewer #2: The manuscript is not written in good English. There are many grammatical, topological and convention related errors some of which are mentioned below.

Comments

Minor:

1. Title is incomplete. Title must include the organism worked upon.

Response: We have changed the title to “ Inhibitory activity of traditional plants against Mycobacterium smegmatis and their action on Filamenting temperature sensitive mutant Z (FtsZ) - a cell division protein”. 

2. Abstract: Do not include the objective as first line of abstract. It should not start with “To…..”

Response: Changes were made in the text

3. Abbreviate the name of the organism/plant wherever its used for the first time in the text.

Response: Changes were made in the text

4. Units used in the text are not constant. Some values are represented in mg/ml, ug/ml and some in uM. It is very difficult for the readers to understand and compare.

Response: Values are converted to mg/mL, ug/mL throughout the manuscript

5. Line 22: Hexane and not ‘hexame’

Response: Change was made in the text

6. Line 39: in vitro should be italicized here as well as throughout the text.

Response: In vitro has changed to In vitro

7. Line 59: H37RV is incorrect throughout the text. Correct as H37Rv.

Response: H37RV is corrected as H37Rv throughout the document.

 8. Section 2.1 says 7H9 broth was procured from Sigma and section 2.5 says the same is from BD. Please correct.

Response: 7H9 broth was procured from Sigma and was changed in the text file.

9. Line 269: Correct the gene name as 16S rRNA

Response: The gene name is corrected as 16S rRNA

10. Line 269: Write Mycobacterium sp. instead of only genus.

Response: M.tuberculosis genus is added

11. Line 283: ‘strain of existing drug’ does not make sense

Response: The sentence were corrected.

12. Line 297: Write observation instead of statement

Response: Change was made in the statement.

13. Line 323: Remove the author’s name for reference 51. Just use number.

Response: We have removed author name.

Major:

14. Section 2.10:

a) If you added plant extract samples after adding 100ul broth, it will change the volume. How did you keep the volume constant?

Response: Microplate serial broth-dilution technique was used according to the below reference. Briefly twice the required maximum concentration of the plant extract is added initially, then it is further serially diluted with broth to reach a minimum concentration. To this 100ul of 0.1 O.D culture was added. Thus the volume is kept constant

b) Why did you use 1mg/ml INH and RIF and not test other concentrations?

Response: The stock of RIF and INH is 1mg/mL of DMSO, from which further 10X dilution was made to avoid DMSO effect .100 µl from the stock concentration was made up to 1mL with the corresponding DMSO/ water. Such that maximum Concentration of 100 µg/mL was used for the assay which is further serially diluted to obtain the minimum concentration that shows inhibition of growth. We have corrected the above statement under material sections.

c) Did you use medium control without DMSO as a control for INH?

Response: Yes, appropriate controls were followed. Only broth as medium control and DMSO also a control 

d) All volumes used should be fixed. Do not write about 20ul and so.

Response: We corrected the statement

e). Why did you use 1X MIC for time kill assay and 1.5X for gene expression studies. Give reference or explanation for selecting these numbers.

Response: 

• 1x MIC was used for time kill based on the synergy studies. 

Explanation: In time-kill the concentrations of plant extracts were used at 1x MIC whereas the concentration of drugs (RIF & INH) was used at sub-minimal inhibitory concentration which was found to be effective in synergy. So, subminimal concentration was used.

• In gene expression studies a higher fold concentrations were employed, since gene expression studies are generally done in lesser time span. Thus 1.5x MIC was considered.

f). Section 3.2, Line 307: ‘hexane wash of plants enhances the activity’..did you study unwashed extracts to compare.

Response: It should be written as Hexane methanol wash to enhance the activity. We didn’t study the unwashed extracts to compare the efficiency. We corrected the statement in the text file

g). Section 3.3: Paragraph 1 results can be represented as a small table for an easy comparison.

Response: Table is included as a separate file.

i). Conclusion: Include references for mode of action of INH and RIF (line 442-444).

Response: References were included for INH and RIF action.

Reviewer #3: 

1, The phytochemistry that was done to determine the presence and quantity of pinitol is not clear. 

Response: We have done both GCMS and HPLC for all the three HXM extracts. From HPLC, results of standard D- Pinitol with a retention time at 10.198 min which matches exactly with the retention value of three plants namely A.nilotica, A.marmelos and G.glabra at 10.187, 10.143 and 10.135 respectively confirming the presence of D-Pinitol. Moreover, mass fragmentation was done for major peak of A.nilotica at 5.162 min retension time and G.glabra at 5.169 min retension time which showed 90% similarity index for D- Pinitol. A.marmelos at 5.160 min retension time showed 92% of similarity index . This major peak of all the three HXM extracts matches with the derivatives of D- Pinitol, pentakis(trimethylsilyl) ether of about 90 %. MS fragmentation reports were also attached in the supplementary file.

The paper is lacking any statistical analysis, although this is mentioned in the M&M section.

Response: We included statistical analysis in the data including p value in all the figures as well as one way Anova.

Response for Line 147: Yes, we dried the plant samples in an oven overnight at room temperature (37οC). A oven with a fan is used, which blew air through the trays.

Response for Line 148: Fruit pulp was macerated by using mortar and pestle without using any solvents. 

Response for Line 166: Stock solution was prepared 1mg/mL

Response for line 175: The concentration of D- Pinitol to run HPLC is 1mg/mL. 

 In the whole text, values were converted in to mg and µg/mL.

Response for line 183: Protocol has been changed

Response for line 190: our plant extracts were dissolved at 100mg/mL of 5 % DMSO. We didn’t observe any microprecipitation because the extract dissolved completely.

Response for line 191: 

The mycobacterium strain was grown in 7H9 medium to an OD of 0.6 and then diluted to attain a cell concentration of 1x106 CFU/ml and added to each well.

Response for line 197: Change has been made

Response for line 224: Centrifugation done at 10,000 rpm which corresponds to 11200g 

Response for line 249: Magnification was done at 40X

Response for line 260: HXM extract/ D- Pinitol conc is 1.5 x MIC 

Response for line 262: Malachite green was added as a color reagent

Response for line 288: Fragmentation patterns with their similarity index were attached in supplementary file.

Response for line 294-296:. Mass spectra data is attached in supplementary file to confirm the presence of D-Pinitol with a similarity index of 90 % in all the three HXM extracts.

Response for line 307: It should be written as hexane methanol wash of the plants enhances the activity. (Changes were made in the text).

Response for line 310: Values are converted to mg and µg/mL.

Response for line 312: Statement should be written as hexane methanol wash of the plant enhances its activity. However, the cited reference states the increase in the activity of hexane extract of A.marmelos this is may be due to variation of plant sources, method of preparation of extracts, preparation of media and growth condition of organism. Cited paper used M.tuberculois for their experiment. All these factors may influence the MIC values of the extracts.

Response for line 317: The reported value is not inhibition, so we deleted the statement in our text.

Response for line 318: We modified the text. 

Reviewer 3 : if D – Pinitol is the major constituent in all the 3 plant extracts and show an low /miC of 0.58um(lower than positive control). surely, the extracts will show much better activity.

Response for line 346: Yes, we agree with your statement. During combinatorial studies, plant extracts at subminimal concentration i.e., below their MIC values were able to reduce the amount of INH and RIF to 0.01µg/ml to achieve good activity.

Reviewer 3 : This is low activity (A. nilotica, A. marmelos and G. glabra inhibited the M.tb FtsZ GTPase activity with an 415 IC50 of 22.50 ± 1.25, 23.45 ± 1.12 and 37.9 ± 1.19 mg/ml respectively). For a plant extract to be activity it should fall within ug/ml conc range especially for enzymatic assays and kinetics. 

Response for line 415 : It should be written as IC50 of 1.399, 1.329, 1.564 mg/mL. All biological activity in this paper was performed in mg level and the concentration level used for this experiment is 0.5 to 2.5mg/mL. So it is expected to fall within mg/mL. Since, D- Pinitol is a phytochemical it expressed GTPase activity in µg/mL. Our study is aimed to design a novel plant inhibitor that reduces the mycobacterial resistance to the other first line drugs.

Reviewer 3 : if the treatment is done at 1.5 times the MIC is expected to see a reduction. An effect should rather be observed at ½, ¼ or 1/8 MIC.

Response : In gene expression studies a higher fold concentrations were employed, since gene expression studies are generally done in lesser time span (4 hr). Thus 1.5x MIC was considered for this study. Whereas MIC studies were performed after 48 hrs of incubation.

Figure 3 : Standard deviations were included.

Figure 4 : Stastical values were added

Figure 5: Stastical values were added and length is changed to µm.

Figure 6: IC 50 values were not calculated for light scattering assay.

---

## [Decision Letter · Decision Letter 1]

28 Jan 2020

PONE-D-19-29098R1

Inhibitory activity of traditional plants against Mycobacterium smegmatis and their action on Filamenting temperature sensitive mutant Z (FtsZ) - a cell division protein

PLOS ONE

Dear Dr Ravindran,

Thank you for submitting your manuscript to PLOS ONE. After careful consideration, we feel that it has merit but does not fully meet PLOS ONE’s publication criteria as it currently stands. Therefore, we invite you to submit a revised version of the manuscript that addresses the points raised during the review process.

ACADEMIC EDITOR: The authors need to carefully revise the manuscript for language, grammar and syntax errors.  Also, the response to reviewers' comments should be included in the manuscript.  Some of the limitations pointed out in the first round of review should be explicitly mentioned in the discussion section. (for example, reviewer#3 mentioned that the activity was weak since it is in the range of mg/ml rather than ng/ml.)

We would appreciate receiving your revised manuscript by Mar 13 2020 11:59PM. To enhance the reproducibility of your results, we recommend that if applicable you deposit your laboratory protocols in protocols.io, where a protocol can be assigned its own identifier (DOI) such that it can be cited independently in the future. For instructions see: http://journals.plos.org/plosone/s/submission-guidelines#loc-laboratory-protocols

We look forward to receiving your revised manuscript.

Kind regards,

Selvakumar Subbian, Ph.D.

Academic Editor

PLOS ONE

Reviewers' comments:

Reviewer's Responses to Questions

**Comments to the Author**

1. If the authors have adequately addressed your comments raised in a previous round of review and you feel that this manuscript is now acceptable for publication, you may indicate that here to bypass the “Comments to the Author” section, enter your conflict of interest statement in the “Confidential to Editor” section, and submit your "Accept" recommendation.

Reviewer #1: (No Response)

Reviewer #2: All comments have been addressed

2. Is the manuscript technically sound, and do the data support the conclusions?

Reviewer #1: Partly

Reviewer #2: Yes

3. Has the statistical analysis been performed appropriately and rigorously? 

Reviewer #1: No

Reviewer #2: Yes

4. Have the authors made all data underlying the findings in their manuscript fully available?

Reviewer #1: No

Reviewer #2: Yes

5. Is the manuscript presented in an intelligible fashion and written in standard English?

Reviewer #1: Yes

Reviewer #2: No

6. Review Comments to the Author

Reviewer #1: Although the manuscript titled “Inhibitory activity of traditional plants against Mycobacterium smegmatis and their action on Filamenting temperature sensitive mutant Z (FtsZ) - a cell division protein” is interesting it is not acceptable for publication in the present version because unless the following concerns are addressed.

Major concerns -

1. Section 3.3 and 3.4 - The combinatorial effect is performed at 24-hour timepoint in the present study using resazurin assay. But in time kill studies with usage of 1x MIC of drugs/extracts, no difference is seen between the tested samples at 24 hours. No representative figures given for the data of HXM extracts in time kill studies though log differences are mentioned in the text in results section. Also, combination of the drugs and extracts have much less reduction when compared to the extracts or pinitol alone. What is the possible explanation?

2. In the cell elongation studies (section 3.6), the cell length cannot be given as a mean. It would be preferable to count atleast 100 cells and give a graph of different cell lengths as percentage. 2- 3 um, 4-5 um, so on. That would be a better representative of the study of what percentage is increase in cell length rather than a single value.

3. In the section 3.7, IC50 value is calculated from linear points and graph for standard compounds looks neither linear or as a dose response. Why were the concentrations chosen as given in the text? They are neither log scale increase nor times MIC. What is the possible reason the authors chose to test them? Is curve fitting done for statistical analysis?

4. The reason for performing gene expression studies at 1.5X MIC is not scientific. If the authors felt 4 hours is too early then the studies should have been done proper titration of time and x MIC. Otherwise there should be a valid reason authors felt that using 1.5x MIC at 4-hour time point is acceptable. A single point data with no scientific explanation to substantiate the same is not convincing.

Reviewer #2: The manuscript still has many grammatical and topological errors. Authors should pay attention to sentence formation as well. I strongly recommend that the authors get the manuscript reviewed by an expert before resubmission.

Comments:

Isoniazid and Rifampicin can be abbreviated as INH and RIF respectively throughout the text.

Line 30: Correct as ‘Time kill kinetic studies indicate..’

Line 31: ‘reduction’ would be better than ‘retardation’

Line 41: ‘in-vitro’ is incorrect. Write ‘in vitro’.

Line 41: inhibition of FtsZ...

Line 134-135: Represent in units and not in weight or volume. Also, mention the method of sterlization for OADC.

Line 139-141: Authors probably want to say that ‘they’ did not need any specific permission for collection. It can be written as ‘No specific permission was required for collection of plant parts....’

Line 143: word ‘species’ is repeated, please remove from the end.

Line 151: Were the dried leaves powdered or crushed? Pls mention in the text.

Line 155, 195, 212, 216: Do not write ‘about’ for fixed volumes.

Line 157: How is the mixture undisturbed if it was stirred? Where was it incubated overnight and at what temperature? Pls correct. Write the temperature used for following steps as well.

Line 162: Already mentioned in section 1.

Line 177: Correct as ‘elution was carried out at ….’

Line 182: Correct the sentence.

Line 193: Write ‘incubated’ instead of ‘kept’.

Line 195: Mention whether 96 well plates used were U-bottom.

line 197: Mention the range of tested concentrations of 3 plant extracts.

Line 203: 48 hours..

Line 204: ‘resazurin dissolved in water’

Line 204: ‘after this incubation’ is repeated in the same sentence

Line 215: Mention which cultures..

Line 217: Correct as ‘plates were further incubated’.

Section 2.10 says that the MIC plates were incubated for 48 hrs while in section 2.11 incubation time is mentioned as 24 hrs. Why is it different for section 2.11? Pls explain.

Line 228: Correct as ‘withdrawn at 0, 1, 2, 3, 4, 5, 6 and 7 days, diluted serially and...’

Line 229: Mention time of incubation for the plates.

Line 235: Have you used ‘they’ for ‘pellet’. Pls correct.

Line 343: The concentrations of plant extracts used are exact MICs. How are these sub-minimal?

Line 375-381: Where are these results shown in the manuscript? These should be represented as a graph and figure number should be mentioned here.

Line 401: Pls explain this estimation.

Line 419: Remove ‘an’.

7. PLOS authors have the option to publish the peer review history of their article (what does this mean?). If published, this will include your full peer review and any attached files.

Reviewer #1: Yes: Radha Gopalaswamy

Reviewer #2: No

---

## [Author Response · Author response to Decision Letter 1]

13 Feb 2020

Response to the comments

Reviewer #1: Although the manuscript titled “Inhibitory activity of traditional plants against Mycobacterium smegmatis and their action on Filamenting temperature sensitive mutant Z (FtsZ) - a cell division protein” is interesting it is not acceptable for publication in the present version because unless the following concerns are addressed.

Major concerns –

1. Section 3.3 and 3.4 – 

Comments : The combinatorial effect is performed at 24-hour time point in the present study using resazurin assay. But in time kill studies with usage of 1x MIC of drugs/extracts, no difference is seen between the tested samples at 24 hours. 

Response: The values of CFU at 24th hour are represented in the table 1. CFU values for combination therapy is lesser than the individual drug/ plant extract. The drugs INH and RIF are bactericidal in nature ( Ref 1). Mycobacterium smegmatis doubling time is 6-8 hours (Ref 2). So beyond 24 hours, drug or extract are not able to prevent the growth of M.smegmatis. It is difficult to differentiate the effect of various treatments after 24 hours in the time kill.

Table 1

Treatment CFU - 24th hr

Control 614± 3.35

INH 522± 4.87

RIF 502± 3.51

A.nilotica 456±1.89

A.nilotica/INH 416±2.32

A.nilotica/RIF 401±3.21

A.marmelos 445±1.91

A.marmelos/INH 404±1.82

A.marmelos/RIF 411±2.67

G.glabra 434±3.76

G.glabra /INH 391± 4.67

G.glabra /RIF 386±4.56

D-Pinitol 401±1.96

D-Pinitol/ INH 294±1.75

D-Pinitol/RIF 302±2.25

 Comments: No representative figures given for the data of HXM extracts in time kill studies though log differences are mentioned in the text in results section.

Response: Representative figures are given in the supplementary data.

Comments: Combination of the drugs and extracts have much less reduction when compared to the extracts or pinitol alone. What is the possible explanation?

Response: 

Isoniazid and Rifampicin are bactericidal in nature (Ref 1) whereas D-Pinitol and the plant extracts are found to be bacteriostatic. Hence in combination, i.e. plant extract or D- Pinitol in higher concentration and drugs in less concentration might alter the effect. This, variation could be observed significantly after 4 days in the time- kill graphs.

The above same effect was observed for D- Pinitol/ Pencillin combination against MRSA. In which they observed, D- Pinitol in combination with Pencillin, showed less activity when compared with D- Pinitol at 1X MIC (Ref 3).

The observed decrease in the activity when HXM plant extracts/ D- Pinitol and drugs are combined together could be due to formation (Mailliard reaction) of adduct between INH/RIF through the amine group with D-Pinitol (sugar). This Mailliard reaction between reducing sugars and amines were reported in studies (Ref 4).

Similar activity observed in plant extracts also. Whereas when D-Pinitol used alone it showed more activity than in combination with other extracts.

However decrease in potency of antibiotics was observed, particularly when combined with A.nilotica/ INH, A.marmelos/ INH and A.marmelos/ RIF. It was particularly apparent for G.glabra, which was able to inhibit the growth of M.smegmatis, but enhanced its activity when combined with anti-tuberculosis drugs. (INH/RIF) (Supplementary Fig 11&12S).

2. In the cell elongation studies (section 3.6), It would be preferable to count atleast 100 cells and give a graph of different cell lengths as percentage. 2- 3 um, 4-5 um, so on. That would be a better representative of the study of what percentage is increase in cell length rather than a single value.

Response: Graph with percentage of cells with different length is provided in Figure 5h.

3. In the section 3.7, IC50 value is calculated from linear points and graph for standard compounds looks neither linear or as a dose response. Why were the concentrations chosen as given in the text? They are neither log scale increase nor times MIC. What is the possible reason the authors chose to test them? Is curve fitting done for statistical analysis?

Response: Yes, The Curve fitting is done and it is represented in Figure 6 a, b. It is done using Graph pad prism. Non linear regression (four parameter model) was performed to obtain the IC50 of the respective treatments. 

4. The reason for performing gene expression studies at 1.5X MIC is not scientific. If the authors felt 4 hours is too early then the studies should have been done proper titration of time and x MIC. Otherwise there should be a valid reason authors felt that using 1.5x MIC at 4-hour time point is acceptable. A single point data with no scientific explanation to substantiate the same is not convincing.

Response: The effect of cell elongation was observed at 4 hr. Hence, we performed the expression studies also at 4 hr. There are studies in which gene expression is performed in response to 90 min, 3 hr and 5 hr exposure also. For eg: Gene expression is done for M.tuberculosis after 3 hr of exposure (Ref 6) 

References: 

1. Bacteriostatic and Bactericidal Activity of Antituberculosis Drugs Against Mycobacterium Tuberculosis, Mycobacterium avium-Mycobacterium Intracellulare Complex and Mycobacterium Kansasii in Different Growth Phases, , S Yamori 1, S Ichiyama, K Shimokata, M Tsukamura, Microbiol Immunol, 36 (4), 361-8, 1992.

2. Adaptation of Mycobacterium smegmatis to Stationary Phase, MARJAN J. SMEULDERS, JACQUIE KEER, RICHARD A. SPEIGHT,† AND HUW D. WILLIAMS, JOURNAL OF BACTERIOLOGY, p. 270–283 Vol(181): 1, 1999.

3. Synergistic effect of (+)-pinitol from Saraca asoca with β-lactam antibiotics and studies on the in silico possible mechanism, Furkan Ahmad, and Laxminarain Misra, Journal of Asian Natural Products Research 18(2):1-12(2015).

4. Maillard Reactions in Pharmaceutical Formulations and Human Health, . Int J Pharm Compd, David W Newton, ,,15 (1), 32-40, 2011

5. Differential Gene Expression in Auristatin PHE-Treated Cryptococcus neoformans Tanja Woyke, Michael E. Berens, Dominique B. Hoelzinger, George R. Pettit, Günther Winkelmann, Robin K. Pettit, ANTIMICROBIAL AGENTS AND CHEMOTHERAPY, , p. 561 567(2004) (Gene expression done at 1.5 x MIC in response to 90 min).

6. Thiol reductive stress induces cellulose-anchored biofilm formation in Mycobacterium tuberculosis,Abhishek Trivedi, Parminder Singh Mavi, Deepak Bhatt & Ashwani Kumar , Nature Communications vol 7: 11392, (2016) .(Gene expression done in response to 3 hr)

Reviewer #2: The manuscript still has many grammatical and topological errors. Authors should pay attention to sentence formation as well. I strongly recommend that the authors get the manuscript reviewed by an expert before resubmission.

Comments:

Isoniazid and Rifampicin can be abbreviated as INH and RIF respectively throughout the text.

Response: Isoniazid and Rifampicin is abbreviated as INH and RIF throughout the text.

Line 30: Correct as ‘Time kill kinetic studies indicate..’

Response: Corrected as Indicated.

Line 31: ‘reduction’ would be better than ‘retardation’

Response: Replaced the word.

Line 41: ‘in-vitro’ is incorrect. Write ‘in vitro’.

Response: Corrected as in vitro

Line 41: inhibition of FtsZ...

Response: inhibition of FtsZ is added

Line 134-135: Represent in units and not in weight or volume. Also, mention the method of sterlization for OADC.

Response: The mixture was filter sterilized using 0.44µm membrane. It was added separately to the culture medium at 2% concentration (v/v). 

Line 139-141: Authors probably want to say that ‘they’ did not need any specific permission for collection. It can be written as ‘No specific permission was required for collection of plant parts....’

Response: Sentences were corrected

Line 143: word ‘species’ is repeated, please remove from the end.

Response: the word species is removed.

Line 151: Were the dried leaves powdered or crushed? Pls mention in the text.

Response: It is mentioned as fine powder.

Line 155, 195, 212, 216: Do not write ‘about’ for fixed volumes.

Response: About is removed

Line 157: How is the mixture undisturbed if it was stirred? Where was it incubated overnight and at what temperature? Pls correct. Write the temperature used for following steps as well.

Response: Sentences were corrected.

Line 162: Already mentioned in section 1.

Response: Line was removed

Line 177: Correct as ‘elution was carried out at ….’

Response: Sentence has been corrected

Line 182: Correct the sentence.

Response: Sentence has been corrected.

Line 193: Write ‘incubated’ instead of ‘kept’.

Response: Replaced the word

Line 195: Mention whether 96 well plates used were U-bottom.

Response: 96 well plates used were flat bottom

line 197: Mention the range of tested concentrations of 3 plant extracts.

Response: The range of tested concentration is 0.75 to 100 mg/mL

Line 203: 48 hours.

Response: corrected as 48 hours.

Line 204: ‘resazurin dissolved in water’

Response: Sentence was corrected

Line 204: ‘after this incubation’ is repeated in the same sentence

Response: after this incubation’ is removed

Line 215: Mention which cultures.

Response: M.smegmatis is added

Line 217: Correct as ‘plates were further incubated’.

Response: The sentence was corrected.

Section 2.10 says that the MIC plates were incubated for 48 hrs while in section 2.11 incubation time is mentioned as 24 hrs. Why is it different for section 2.11? Pls explain.

Response: MIC was done at 48 hrs incubation. While, we observed synergy at 24 hrs, since a combination showed better action and it was done at 24 hrs.

Line 228: Correct as ‘withdrawn at 0, 1, 2, 3, 4, 5, 6 and 7 days, diluted serially and...’

Response: The sentence was corrected.

Line 229: Mention time of incubation for the plates.

Response : Time of incubation is 48 hours

Line 235: Have you used ‘they’ for ‘pellet’. Pls correct.

Response: The word pellet is added .

Line 343: The concentrations of plant extracts used are exact MICs. How are these sub-minimal?

Response: In synergistic studies, plant extracts acted in subminimal concentration along with the combination of INH/ RIF by reducing the MIC of first line drugs from 4.00 to 0.01 μg/ mL for INH and 2.00 to 0.01 μg/ m for RIF.

Line 375-381: Where are these results shown in the manuscript? These should be represented as a graph and figure number should be mentioned here.

Response: Figure number is included in the manuscript

Line 401: Pls explain this estimation.

Response : The sentences were corrected. 

Line 419: Remove ‘an’.

Response: The word “an” is removed.

---

## [Decision Letter · Decision Letter 2]

3 Apr 2020

PONE-D-19-29098R2

Inhibitory activity of traditional plants against Mycobacterium smegmatis and their action on Filamenting temperature sensitive mutant Z (FtsZ) - a cell division protein

PLOS ONE

Dear Dr Ravindran,

Thank you for submitting your manuscript to PLOS ONE. After careful consideration, we feel that it has merit but does not fully meet PLOS ONE’s publication criteria as it currently stands. Therefore, we invite you to submit a revised version of the manuscript that addresses the points raised during the review process.

Major comments:

1. The response to the reviewer’s concerns should be inserted appropriately in the main text of the revised version.  It is not enough to present only in the rebuttal letter. For example, add “Mycobacterium smegmatis doubling time is 6-8 hours; So beyond 24 hours, drug or extract is not able to prevent the growth of M.smegmatis.” as you mentioned in the rebuttal letter

2. The main component in the plant extracts tested, was deemed to be D-Pinitol, based on RT with the standard. However, no direct evidence has been shown to prove that the component is D-Pinitol.  This limitation should be mentioned in line #367.

3. There is confusion in Table-1. While in the main manuscript, Table-1 is showing MIC values, in the “response to reviewer” document, Table-1 shows CFU data (not sure if this is log scale or linear). This requires clarification and fixing. Both tables should be presented in the main manuscript.

4. The data presented in Supplementary table-1 is unclear as to what it represents. The heading should appear as the caption at the top of the table. Plus, the RT and % area seems to be misplaced.  Also, I see peak#15 and 17 as D-pinitol, with different RT and %area.  Same discrepancy between HPLC and GC of each of the plant extracts. These are not consistent with the numbers in the main text (line#312). How do the RT values in the suppl tables of GCMS, correlate with the respective spectrum shown in Suppl-figs-2 to 4 ?. These entities need a full, detailed explanation in the manuscript.

5. Section 3.5. mentions that about 50% of cells have “Cell wall damage and presumably cell death”.  How does this observation correlate with the “Cell elongation study”, where it is argued that after treatment with extracts, the cells elongate and fail to replicate and undergo death ?.

6. Based on the above point, if M. smeg is killed by cell wall damage and cell elongation during treatment with plant extracts, then how would you justify the increase in CFU of treated bacteria in Fig-3. ?

7. The justification to use 1.5x Mic and 4hrs post treatment for gene expression studies should be mentioned in the methods or results section. Mere citation is insufficient, as this particular study and the organism/conditions used are unique and presumably not presented elsewhere in the literature.

Minor comments:

Lines 369-371. Should be rephrased or removed, since it is speculative. The current study is performed in M. smeg, which is not the causative agent of tuberculosis in humans.

In the rebuttal letter, Milliard reaction should be corrected to Millard reaction, and the justification should be added in the main manuscript.3e

We would appreciate receiving your revised manuscript by May 18 2020 11:59PM. To enhance the reproducibility of your results, we recommend that if applicable you deposit your laboratory protocols in protocols.io, where a protocol can be assigned its own identifier (DOI) such that it can be cited independently in the future. For instructions see: http://journals.plos.org/plosone/s/submission-guidelines#loc-laboratory-protocols

We look forward to receiving your revised manuscript.

Kind regards,

Selvakumar Subbian, Ph.D.

Academic Editor

PLOS ONE

Reviewers' comments:

Reviewer's Responses to Questions

**Comments to the Author**

1. If the authors have adequately addressed your comments raised in a previous round of review and you feel that this manuscript is now acceptable for publication, you may indicate that here to bypass the “Comments to the Author” section, enter your conflict of interest statement in the “Confidential to Editor” section, and submit your "Accept" recommendation.

Reviewer #2: All comments have been addressed

2. Is the manuscript technically sound, and do the data support the conclusions?

Reviewer #2: Yes

3. Has the statistical analysis been performed appropriately and rigorously? 

Reviewer #2: Yes

4. Have the authors made all data underlying the findings in their manuscript fully available?

Reviewer #2: Yes

5. Is the manuscript presented in an intelligible fashion and written in standard English?

Reviewer #2: Yes

6. Review Comments to the Author

Reviewer #2: (No Response)

7. PLOS authors have the option to publish the peer review history of their article (what does this mean?). If published, this will include your full peer review and any attached files.

Reviewer #2: No

---

## [Author Response · Author response to Decision Letter 2]

14 Apr 2020

Major comments:

1. The response to the reviewer’s concerns should be inserted appropriately in the main text of the revised version. It is not enough to present only in the rebuttal letter. For example, add “Mycobacterium smegmatis doubling time is 6-8 hours; So beyond 24 hours, drug or extract is not able to prevent the growth of M.smegmatis.” as you mentioned in the rebuttal letter.

Response: The above mentioned lines were added in the manuscript. 

2. The main component in the plant extracts tested, was deemed to be D-Pinitol, based on RT with the standard. However, no direct evidence has been shown to prove that the component is D-Pinitol. This limitation should be mentioned in line #367.

Response: 

• D-Pinitol detection is based on RT in HPLC with the standard. This is now mentioned in this manuscript.

• Also, GC-MS Mass fragmentation was done for peak of A.nilotica at 5.162 min retension time and G.glabra at 5.169 min retention time which showed 90% similarity index for D- Pinitol. A.marmelos at 5.160 min retention time showed 92% of similarity index .

• HPLC on st andard showed percentage area occupied by D- Pinitol at Retention Time (RT) 10.198 is 100 %.

• HPLC on the three plant HXM extracts A.nilotica, A.marmelos and G.glabra showed percentage area occupied as 45.12 %, 48.33 % and 55.02 % at RT 10.187, 10.143 and 10.135 respectively which matches very closely with RT of standard D-Pinitol (10.198). (Table 1).

• HPLC – matching of retention time is considered as one of the strongest evidence. 

• So based on all these analysis it is concluded the HXM plant extracts has D-Pinitol (Supplementary Figures 7S,8S,9S,10S).

 Table 1: GCMS and HPLCanalysis of D-Pinitol with the retention time and percentage area of the plant extracts (We included this table in main manuscript)

S.NO Name Retension time % Area

 GCMS

1 A.nilotica 5.162 31.83

2 A.marmelos 5.160 5.86

3 G.glabra 5.169 10.98

 HPLC

4 Standard D-Pinitol 10.198 100.00

5 A.nilotica 10.187 45.12

6 A.marmelos 10.143 48.33

7 G.glabra 10.135 55.02

3. There is confusion in Table-1. While in the main manuscript, Table-1 is showing MIC values, in the “response to reviewer” document, Table-1 shows CFU data (not sure if this is log scale or linear). This requires clarification and fixing. Both tables should be presented in the main manuscript.

Response: In the manuscript we have now included the CFU data (linear) as Table-3. 

4. The data presented in Supplementary table-1 is unclear as to what it represents. The heading should appear as the caption at the top of the table. Plus, the RT and % area seems to be misplaced. Also, I see peak#15 and 17 as D-pinitol, with different RT and % area. Same discrepancy between HPLC and GC of each of the plant extracts. These are not consistent with the numbers in the main text (line#312). How do the RT values in the suppl tables of GCMS, correlate with the respective spectrum shown in Suppl-figs-2 to 4 ?. These entities need a full, explanations that should be mentioned in the manuscript.

Response: 

• The headings are changed to the top of the table. I have checked and RT and % area are not misplaced. 

• Peaks 15 shows RT at 4.958 with 10.94 % area and it indicates the presence of D-Pinitol derivatives and its esters with different molecular weights (GCMS library database). Peak 17 of A.nilotica shows D-Pinitol at RT 5.162 with 31.83 % area. RT of A.nilotica matches with the RT of A.marmelos and G.glabra. 

• The numbers in the main text (line#312) correspond to the RT of D-Pinitol in HPLC reports and they are consistent with the corresponding reports in supplementary figures 7S,8S and 9S.

• To better show the correlation between the spectrum and the respective table, we have now included the original GCMS report as a figure that has both the spectrum as well as the table (Supplementary figures 1S,2S and 3S).

• Explanations on GCMS and HPLC peaks were included in the main manuscript with a table.

5. Section 3.5. mentions that about 50% of cells have “Cell wall damage and presumably cell death”. How does this observation correlate with the “Cell elongation study”, where it is argued that after treatment with extracts, the cells elongate and fail to replicate and undergo death ?.

Response: 

• The cell elongation histogram (Figure 5h) shows the percentage of elongated cells, from this it can be observed that more than 50 % of cells were elongated. Comparing these percentages with the cell wall damage percentages, it can be derived that there is a percentage of cells that are both elongated and have cell wall damage. Antibacterial compounds can have different modes of action. Cell wall damage is another mechanism of cell death and there need not be a correlation between cell elongation and cell wall damage.

6. Based on the above point, if M. smeg is killed by cell wall damage and cell elongation during treatment with plant extracts, then how would you justify the increase in CFU of treated bacteria in Fig-3. ?

Response: 

• Minimum inhibitory concentration (MIC) is defined as the lowest drug concentration that shows 90% growth inhibition of M. smegmatis (Ref1 and Ref2). The assay adopted in our research also measures this MIC 90

• In our studies, time kill assay, cell elongation and cellular damage studies were carried out at 1x MIC. At 1x MIC, (as per the definition of MIC) 10 % of the organisms which survive at this specific concentration will acclimatize to this conditions and replicates at further time intervals. Thus, an increase in the CFU of the organism is observed in the graph as a function of time (Figure 3a,b). The same observation has been made in other literature reports as mentioned below:

• γ Borono Phosphonate Compounds inhibited 90% of M smegmatis (Ref 3).

• Using Rifampicin at 1 × MIC90 , 10% of plated bacteria which survived, started to grow. With a fractional increase in drug concentration (1.2 × MIC90), the number of survivor colonies dramatically fell and growth of colonies were undetectable at 1.4 × MIC90 (Ref 4).

• Higher concentration of extracts were not used in our study since the objective of this study is to find the efficacy of the plant extracts and drugs combinations at lower concentration to inhibit the action of M.smegmatis.

• We have included the definition of MIC and the cause for increase in CFU in the manuscript.

References:

1) P.Fyhrquist,I. Laakso,S. Garcia Marco,R.Julkunen-Tiitto,R.Hiltunen, (2014)Antimycobacterial activity of ellagitannin and ellagic acid derivate rich crude extracts and fractions of five selected species of Terminalia used for treatment of infectious diseases in African traditional medicine, South African Journal of Botany, (90), 1-16.

2) Akiho Yagi, Ryuji Uchida, Hiroshi Hamamoto, Kazuhisa Sekimizu, Ken-ichi Kimura, Hiroshi Tomod (2017), Anti-Mycobacterium activity of microbial peptides in a silkworm infection model with Mycobacterium smegmatis, The Journal of Antibiotics ,( 70), 685–690.

3)Giulia Mancini, Maria Bouda, James M. Gamrat, and John W. Tomsho, (2019), Synthesis and Antimicrobial Evaluation of γ Borono PhosphonateCompounds inEscherichia coliandMycobacterium smegmatis , ACSOmega,(4), 14551−14559.

4) Jun-Hao Zhu, Bi-Wei Wang, Miaomiao Pan, Yu-Na Zeng, Hesper Rego, and Babak Javid ,Rifampicin can induce antibiotic tolerance in mycobacteria via paradoxical changes in rpoB transcription, Nat Commun. 2018; 9: 4218.

7. The justification to use 1.5x MIC and 4hrs post treatment for gene expression studies should be mentioned in the methods or results section. 

Response: In our studies, cell elongation and cell damage of the M.smegmatis is observed at 4 hours of treatment. Therefore gene expression studies is done at 4 hrs with 1.5 x MIC of plant extracts/ compounds in order to increase the expression of FtsZ. Similar to our studies, selected compounds like Piperidine, Bromo di-methoxy coumarin, Di-methyl amino methyl coumarin at the concentration of 1.5 x MIC increased the FtsZ gene expression studies. The justification is now mentioned in section 3.9.

Reference:

1. Kartik Mitra, Anju Chada, Mukesh Doble, (2019),Pharmacophore based approach to screen and evaluate novel Mycobacterium cell division inhibitors targeting FtsZ – A modelling and experimental study, European Journal of Pharmaceutical Sciences, 135(2019)103-112.

Minor comments:

8, Lines 369-371. Should be rephrased or removed, since it is speculative. The current study is performed in M. smeg, which is not the causative agent of tuberculosis in humans. 

Response: Sentence was removed.

9, In the rebuttal letter, Milliard reaction should be corrected to Millard reaction, and the justification should be added in the main manuscript.3e 

Response: Justification is added in the main manuscript

---

## [Editor Report · Decision Letter 3]

16 Apr 2020

Inhibitory activity of traditional plants against Mycobacterium smegmatis and their action on Filamenting temperature sensitive mutant Z (FtsZ) - a cell division protein

PONE-D-19-29098R3

Dear Dr. Ravindran,

We are pleased to inform you that your manuscript has been judged scientifically suitable for publication and will be formally accepted for publication once it complies with all outstanding technical requirements.

With kind regards,

Selvakumar Subbian, Ph.D.

Academic Editor

PLOS ONE
---

## [Editor Report · Acceptance letter]

21 Apr 2020

PONE-D-19-29098R3 

Inhibitory activity of traditional plants against *Mycobacterium smegmatis* and their action on Filamenting temperature sensitive mutant Z (FtsZ) - a cell division protein 

Dear Dr. Ravindran:

I am pleased to inform you that your manuscript has been deemed suitable for publication in PLOS ONE. Congratulations! Your manuscript is now with our production department. 

With kind regards,

on behalf of

Dr. Selvakumar Subbian 

Academic Editor

PLOS ONE